# Descending pathways from the superior colliculus mediating autonomic and respiratory effects associated with orienting behaviour

Erin Lynch[1], Bowen Dempsey[1] (ID), Christine Saleeba[1], Eloise Monteiro[1], Anita Turner[1] (ID), Peter G. R. Burke[1] (ID), Andrew M. Allen[2] (ID), Roger A. L. Dampney[3], Cara M. Hildreth[1] (ID), Jennifer L. Cornish[1] (ID), Ann K. Goodchild[1] (ID) and Simon McMullan[1] (ID)

[1]*Macquarie Medical School, Faculty of Medicine, Health & Human Sciences, Macquarie University, Sydney, New South Wales, Australia*
[2]*Department of Physiology, University of Melbourne, Victoria, Australia*
[3]*School of Medical Sciences (Physiology), University of Sydney, Sydney, New South Wales, Australia*

Handling Editors: Harold Schultz & Vaughan Macefield

The peer review history is available in the Supporting information section of this article (https://doi.org/10.1113/JP283789#support-information-section).

**Abstract** The ability to discriminate competing external stimuli and initiate contextually appropriate behaviours is a key brain function. Neurons in the deep superior colliculus (dSC) integrate multisensory inputs and activate descending projections to premotor pathways responsible for orienting, attention and defence, behaviours which involve adjustments to respiratory and cardio-vascular parameters. However, the neural pathways that subserve the physiological components of orienting are poorly understood. We report that orienting responses to optogenetic dSC stimulation are accompanied by short-latency autonomic, respiratory and electroencephalographic effects in awake rats, closely mimicking those evoked by naturalistic alerting stimuli. Physiological responses

---

This article was first published as a preprint. Lynch E, Dempsey B, Saleeba C, Monteiro E, Turner A, Burke PGR, Allen AM, Dampney RAL, Hildreth CM, Cornish JL, Goodchild AK, Mullan SMc. 2022. Autonomic and respiratory components of orienting behaviors are mediated by descending pathways originating from the superior colliculus. bioRxiv. https://doi.org/10.1101/2021.06.10.447470

The Journal of Physiology

were not accompanied by detectable aversion or fear, and persisted under urethane anaesthesia, indicating independence from emotional stress. Anterograde and trans-synaptic viral tracing identified a monosynaptic pathway that links the dSC to spinally projecting neurons in the medullary gigantocellular reticular nucleus (GiA), a key hub for the coordination of orienting and locomotor behaviours. In urethane-anaesthetized animals, sympathoexcitatory and cardiovascular, but not respiratory, responses to dSC stimulation were replicated by optogenetic stimulation of the dSC–GiA terminals, suggesting a likely role for this pathway in mediating the autonomic components of dSC-mediated responses. Similarly, extracellular recordings from putative GiA sympathetic pre-motor neurons confirmed short-latency excitatory inputs from the dSC. This pathway represents a likely substrate for autonomic components of orienting responses that are mediated by dSC neurons and suggests a mechanism through which physiological and motor components of orienting behaviours may be integrated without the involvement of higher centres that mediate affective components of defensive responses.

(Received 30 August 2022; accepted after revision 14 October 2022; first published online 28 October 2022)

**Corresponding author** Simon McMullan: Macquarie Medical School, Faculty of Medicine, Health & Human Sciences, Macquarie University, 75 Talavera Road, Sydney, NSW 2109, Australia. Email: simon.mcmullan@mq.edu.au

**Abstract figure legend** Salient acoustic and visual stimuli elicit stereotypical responses with motor, respiratory and cardiovascular components. Here we show that optogenetic stimulation of the deep caudolateral superior colliculus (SC) elicits orienting and arousal that is accompanied by tail vasoconstriction and increased respiratory activity in awake rats. Using a combination of electrophysiology in urethane-anaesthetized rats and anterograde and trans-synaptic viral tracing, we reveal a monosynaptic tecto-medullary pathway from the superior colliculus to spinally projecting neurons in the gigantocellular reticular nucleus (GiA) that contributes to the sympathetic and cardiovascular components of SC responses. We propose that this pathway likely mediates components of innate motor and physiological responses to ecologically relevant naturalistic stimuli.

## Key points

- Neurons in the deep superior colliculus (dSC) integrate multimodal sensory signals to elicit context-dependent innate behaviours that are accompanied by stereotypical cardiovascular and respiratory activities.
- The pathways responsible for mediating the physiological components of colliculus-mediated orienting behaviours are unknown. We show that optogenetic dSC stimulation evokes transient orienting, respiratory and autonomic effects in awake rats which persist under urethane anaesthesia.
- Anterograde tracing from the dSC identified projections to spinally projecting neurons in the medullary gigantocellular reticular nucleus (GiA). Stimulation of this pathway recapitulated autonomic effects evoked by stimulation of dSC neurons.
- Electrophysiological recordings from putative GiA sympathetic premotor neurons confirmed short latency excitatory input from dSC neurons.
- This disynaptic dSC–GiA–spinal sympathoexcitatory pathway may underlie autonomic adjustments to salient environmental cues independent of input from higher centres.

**Erin Lynch** conducted the physiological experiments of this study during her PhD candidature at Macquarie University, Sydney, Australia, under the supervision of Simon McMullan. Erin has wide-ranging neuroscience interests and research experience including neuroanatomy, physiology, pharmacology and psychology. She is now based at the Brain and Mind Institute in Sydney and her research currently focuses on understanding the neurobiology and physiological consequences of psychological disorders such as social anxiety, and in developing more effective pharmacological treatments for various psychological disorders.

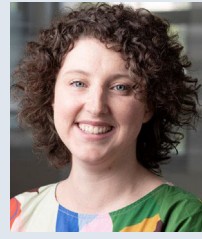

## Introduction

The superior colliculus (SC, optic tectum in non-mammals) is a phylogenetically highly conserved brain region that plays a critical role in generating immediate orienting or defence responses to salient environmental stimuli in vertebrates (Stein & Meredith, 1993; Stein & Stanford, 2008). The SC is a multi-layered structure, consisting of a superficial layer that receives visual inputs and deeper layers that receive convergent inputs from visual, auditory and somatosensory receptors, as well as from other brain regions including the basal ganglia and frontal and parietal cortices (Boehnke & Munoz, 2008; Corneil & Munoz, 2014; Dean et al., 1989; May, 2006; Meredith & Stein, 1986). The majority of multisensory neurons in the deep layers of the SC (dSC) have direct descending projections to motor and premotor nuclei in the brain stem and spinal cord that mediate behavioural responses (Meredith & Stein, 1986). The firing pattern of these multisensory neurons depends upon the pattern of inputs from peripheral receptors and on inputs from other brain regions (Boehnke & Munoz, 2008; Gandhi & Katnani, 2011; Meredith & Stein, 1986). These neurons therefore integrate inputs from multiple sources and generate specific behavioural responses that are appropriate for the particular salient environmental stimulus at any given moment (Boehnke & Munoz, 2008; Dean et al., 1989; Meredith & Stein, 1986).

Apart from orienting and defence behaviours, novel salient environmental stimuli also generate pronounced cardiovascular and respiratory effects. For example, an unexpected sound typically generates an increase in arterial pressure (Baudrie et al., 1997; Caraffa-Braga et al., 1973; Rettig et al., 1986; Yu & Blessing, 1997), mesenteric and cutaneous vasoconstriction (Caraffa-Braga et al., 1973; Yu & Blessing, 1997), and increases in respiratory rate (Kabir et al., 2010; Nalivaiko et al., 2012). Changes in heart rate, however, are generally small and variable (Baudrie et al., 1997; Kabir et al., 2010; Nalivaiko et al., 2012). Measurements of these cardiovascular and respiratory effects are also typically accompanied by cortical electroencephalographic (EEG) desynchronization, an indicator of arousal (Nalivaiko et al., 2012; Yu & Blessing, 1997). As well as auditory stimuli, visual or somatosensory alerting stimuli can also evoke similar cardiovascular and respiratory responses (Kabir et al., 2010; Yu & Blessing, 1997).

There have been extensive studies on the behavioural motor effects generated by stimulation of neurons in the dSC and on the organization and connections of the descending pathways that mediate these motor effects (Dean et al., 1989; Gandhi & Katnani, 2011; Isa et al., 2020; Sahibzada et al., 1986). In contrast, little is known about the neural circuitry that mediates the autonomic and respiratory changes that accompany these behavioural responses. It has been shown that electrical or chemical stimulation of sites within the dSC of anaesthetized rats evokes increases in arterial pressure, respiratory activity and sympathetic vasomotor nerve activity (Iigaya et al., 2012; Keay et al., 1988, 1990). In addition, under conditions where neurons in the dSC are disinhibited, highly synchronized increases in renal sympathetic activity and respiratory activity can be evoked in anaesthetized rats by auditory, visual and somatosensory alerting stimuli (Muller-Ribeiro et al., 2014, 2016). These observations therefore suggest the hypothesis that neurons in the dSC, as well as generating behavioural motor responses to alerting environmental stimuli, also generate autonomic and respiratory changes that are appropriate for the particular motor response.

In this study, using optogenetic stimulation in conscious rats, we show that the orienting behaviour evoked by dSC stimulation is accompanied by short-latency autonomic, respiratory and EEG effects that closely mimic responses evoked by naturalistic alerting stimuli. We also show that these autonomic and respiratory responses occurred in the absence of detectable aversion or fear responses, indicating that they are not a consequence of emotional stress. Finally, anatomical observations and electrophysiological experiments indicate that the physiological effects arising from dSC stimulation are driven by a descending pathway that includes a relay in the ventromedial medulla oblongata.

## Methods

### Ethical approval and animal welfare

Experiments were approved by the Macquarie University Animal Ethics Committee (2014-022 and 2018-024), conformed to the Australian Code of Practice for the Care and Use of Animals for Scientific Purposes 2013, and conformed to the principles and guidelines of *The Journal of Physiology* (Grundy, 2015). Adult rats were housed at the Macquarie University Central Animal Facility in groups of two in individually ventilated cages with environmental enrichment materials. Food and water were available *ad libitum*, cages were kept in a temperature and humidity controlled room (21 ± 2°C, 60% humidity) with a fixed 12 h light/dark cycle.

### Key resource table

| Reagent type/species/resource | Designation | Source, cat # | Additional information |
|---|---|---|---|
| Male Sprague Dawley rats | Sprague Dawley rats | Animal Resources Centre (Perth, WA) | 250–500 g |
| Anaesthetic | Isoflurane | Cenvet | 5% induction, 2–3% maintenance |
| Anaesthetic | Ketamine-medetomidine | Parnell laboratories (ketamine); Pfizer Animal Health (medetomidine) | 75 mg kg$^{-1}$ ketamine + 0.75 mg kg$^{-1}$ medetomidine I.P. |
| Anaesthetic | Urethane | Sigma Aldrich, Australia | 1.3 g kg$^{-1}$ I.P. @ 10% in NaCl |
| Anaesthetic | Lethabarb | Virbac, Australia | 200 mg kg$^{-1}$ in 0.9% saline I.P. (1:4 dilution) for conscious rats. 150 mg kg$^{-1}$ in 0.9% saline I.V. at conclusion of urethane-anaesthetized electrophysiology experiments. |
| Antibiotic | Cephazolin sodium | Mayne Pharma, Australia | 100 mg kg$^{-1}$ I.M. |
| Analgesic | Carprofen | Norbrook Pharmaceuticals, Australia | 5 mg kg$^{-1}$ s.c. |
| Neuromuscular blocker | Pancuronium bromide | Astra Pharmaceuticals, Australia | 0.4 mg induction, 0.2 mg maintenance as required |
| Local anaesthetic | Bupivacaine | Pfizer | 0.25 mg s.c. |
| Anaesthetic reverser | Atipamezole | Pfizer | 1 mg kg$^{-1}$ s.c. (ketamine-medetomidine anaesthetized animals only) |
| Thermal camera | IR camera | FLIR, Sweden, #P45 | 320 × 240 pixel greyscale output converted to tjp15328 8 bit digital video by generic USB video converter (20–36°C range, ±2% accuracy) |
| Viral vector, optogenetic stimulation | AAV2-Syn-ChR2(H134R)-eYFP | Addgene 26973 | 1.79E+12 vg ml$^{-1}$ |
| Viral vector, optogenetic stimulation control | AAVDJ-Syn1-DIO-eGFP & AAVDJ-CMV-Cre | Salk Vector Core | 2.31E+12 vg ml$^{-1}$ (DIO-eGFP), 9.43E+11 (CMV-Cre): mixed 1:1 |
| Viral vector, anterograde tracing | AAV2-CBA-tdTomato | Manufactured in-house | 6.32E+12 vg ml$^{-1}$ |
| Viral vector Cre-tagging | AAV1-hSyn-Cre.WPRE.hGH | Addgene #105553 | 2.6E+13 vg ml$^{-1}$ |
| Viral vector Cre-tagging | AAVretro-hSyn-DIO-EGFP | Addgene #50457 | 7E+12 vg ml$^{-1}$ |
| Axonal tracer | Cholera Toxin B (CTB) | List Biological Laboratories, USA, #106 | 0.5% |
| Primary antibody | Goat anti-CTB | List Biological Laboratories, #703 | 1:1000 |
| Primary antibody | Mouse anti- Tyrosine Hydroxylase | Sigma, #T1299 | 1:1000 |
| Primary antibody | Sheep anti-CHX10 | Millipore, #AB9016 | 1:500 |
| Primary antibody | Chicken anti-GFP | Abcam, #ab13970 | 1:1000 |
| Secondary antibody | Donkey-anti-Chicken 488 | Sigma, #SAB4600031 | 1:500 |
| Secondary antibody | Donkey anti- Goat IgG (H+L chain) Alexa Fluor 647 | Life Technologies, #A21447 | 1:500 |
| Secondary antibody | Donkey anti- Mouse IgG (H+L chain) Alexa Fluor 647 | Life Technologies, #A31571 | 1:500 |

## Animal preparation–recovery experiments

**dSC vector injection and optical fibre implantation.**
Experiments focused on a region of the rat caudolateral dSC (cl-dSC) which, when disinhibited, unmasks respiratory, cardiovascular and somatomotor responses to visual and auditory stimuli under urethane anaesthesia (Muller-Ribeiro et al., 2014, 2016). Rats were anaesthetized with ketamine-medetomidine or isoflurane in oxygen and treated with prophylactic antibiotic, local anaesthetic and analgesic drugs. Anaesthetic depth was monitored by assessing respiratory and motor responses to a firm hind paw pinch. Unilateral injections of viral vectors (200–300 nl) were made at the cl-dSC (2 mm left of midline, 0.5–1 mm rostral, 4.5–5 mm deep to Lambda) using borosilicate pipettes under the control of a NeuroStar stereotaxic robot. Pneumatic injections were made over 10 min and pipettes kept in position for 5 min prior to withdrawal. For technical reasons, dSC injections were limited to the left side of the brain in the final experimental cohort, but pilot experiments included rats instrumented in both sides; equivalent behavioural responses were observed irrespective of side injected.

For cl-dSC stimulation experiments custom-made (Sparta et al., 2012) 200 $\mu$m optical fibres in 2.5 mm ceramic ferrules were inserted 300 $\mu$m dorsal to AAV2-Syn-ChR2(H134R)-eYFP or control vector (AAVDJ-Syn1-DIO-eGFP & AAVDJ-CMV-Cre) injection sites and secured to the skull with dental acrylic. Transmission efficiency was noted prior to implantation and used to calibrate laser output (LSR473NL, Lasever Inc) to yield 10 mW maximal output during 20 ms light pulses, equivalent to 320 mW/mm$^2$. In a cohort of animals, electroencephalographic (EEG) signals were acquired via Teflon-coated silver wires (0.005', #786000, A-M systems) wrapped around stainless steel jewellers' screws (1 mm diameter) that anchored the acrylic cap (Burke et al., 2014) and exteriorized via a 6-pin board–board connector header (SPC20500, Element14.com.au).

**Spinal tracer injection.** Three weeks after dSC injection of AAV-CBA-tdTomato a subset of rats received injections of CTB in the T2 spinal cord at coordinates that correspond with the interomediolateral cell column, as described previously (0.75 mm lateral, 1 mm deep: Turner et al., 2013). The spinal processes of ketamine-medetomidine anaesthetized rats were clamped to maintain the spine in a horizontal and elevated position, the T2 spinal cord was exposed, the dura punctured and two 200 nl injections made on each side over a period of 5 min before the pipette was slowly retracted. The exposed spinal cord was irrigated with sterile physiological saline, covered with oxidized cellulose haemostat, the wound closed and anaesthesia reversed. Post-operative care and monitoring

were as described below and rats were perfused 5 days later.

## Experimental protocol: behavioural experiments

Rats were habituated to fibre ferrule cleaning, handling, and the attachment of fibre-optic patch cables for a minimum of three sessions prior to behavioural testing, starting 1–2 weeks after optrode implantation, and were also habituated to the rooms used for tail thermography (but not those used for open field test (OFT) or elevated plus maze (EPM) trials). Ferrules were cleaned with 80% ethanol and lubricated with WD-40.

The general effects of unilateral dSC stimulation were investigated in an initial cohort of rats using a low-power 470 nm LED system (DC4100 driver with M470F3 light source, ThorLabs, 1–3 mW output). The home cage was moved into a closely fitting high-sided black wooden box, the lid, conspecifics, bedding and environmental enrichment removed, the fibre ferrule cleaned and lubricated, and rats left to habituate for 15 min. A patch cable fitted with a rotary joint was then attached to the fibre-optic ferrule and suspended from an overhead arm and rats were allowed to habituate for a further 15 min. Trains of dSC stimulation were completed at 5, 10 or 20 Hz using 20 ms light pulses. Sessions were recorded by video and qualitatively analysed to categorize the range of responses seen in ChR2 and control rats. Responses were consistent with those reported in response to stimulation of the same region of the dSC in mice, and so were not quantified in detail (Isa et al., 2020).

**Open field test, elevated plus maze and ultrasonic vocalization.** The OFT, EPM and incidence of spontaneous ultrasonic vocalization (USV) were used to assess whether cl-dSC stimulation was associated with correlates of anxiety-like behaviours at intensities that caused motor effects.

OFT experiments were conducted during the late part of the rats' light phase in a room that was novel to rats. The apparatus was a $1 \times 1$ m white box with 40 cm high walls, illuminated overhead ($\sim$400 Lumens). An infrared video camera was mounted overhead to record behaviour (Motmen Tracker, Motion Mensura, Sydney). The fibre ferrule was cleaned and lubricated and the rat was habituated to the room for 15 min in the home cage (open lid, home cage contained in closely fitting high-sided black wooden box) before patch cable attachment. The rat was habituated for a further 15 min prior to cl-dSC stimulation (320 mW/mm$^2$, 1 Hz, 20 ms pulse width, 2 min) after which the patch cable was disconnected, the rat placed in the apparatus, and the operator left the room for the remainder of the 15 min trial. OFT videos were scored

using AnyMaze software (Stoelting). The OFT arena was cleaned with 70% ethanol between trials.

EPM experiments were conducted during the late part of the rats' light phase in a laboratory that was novel to rats using an opaque grey Perspex cross-shaped apparatus raised 65 cm above the floor and illuminated overhead (~400 Lumens). The arms measured 50 cm long and 10 cm wide, with the closed arms surrounded by a 40 cm high wall. An infrared camera suspended above the apparatus recorded animal movement. The EPM was cleaned with 70% ethanol between trials.

The fibre ferrule was cleaned and lubricated and the rat habituated to the laboratory for 15 min in the home cage within a high-walled black wooden box before patch cable attachment and a further 15 min habituation before photostimulation (320 mW/mm², 1 Hz, 20 ms pulses, 1 min) The patch cable was then disconnected and the rat placed on the EPM for 5 min with the operator outside the room. Videos of EPM exploration were scored using AnyMaze software. Parameters measured were time spent in closed and open arms and at the far ends of open arms. The EPM was cleaned with 70% ethanol between trials. USVs were recorded before and during dSC stimulation and during exploration of the EPM using a M500 USB microphone (10–210 kHz, Pettersson Elektronik, Uppsala, Sweden) and sampled at 500 kHz with 16-bit resolution using BatSound Lite (Pettersson Elektronik). Ultrasound recordings were imported to Spike 2 v7.20 (Cambridge Electronic Design, UK) and converted to spectrograms to enable visualization of USVs, identified using criteria described by Wright et al. (2010) and Brudzynski (2013), and categorized as low or high frequency (18–30 and 40–90 kHz, respectively). Analysis was completed blind to treatment groups.

**Plethysmography and electroencephalography.** To assess the effects of dSC stimulation on respiratory frequency and EEG activity, rats were habituated to a custom-built positive-pressure whole-body plethysmography chamber constructed from a 5.4 l glass desiccating jar fitted with an overhead airtight port for passage of cabling. Respiratory frequency was extracted from the peaks of DC-removed, bandpass filtered (0.1–20 Hz) recordings of chamber air pressure, captured using Spike 2 using a Power1401 ADC (Cambridge Electronic Design) (Burke, Kanbar, Basting et al., 2015). EEG was amplified and bandpass filtered (0.1–100 Hz, CME BMA 400) and simultaneously recorded. Stimulation parameters were titrated such that 10 s trains of 5–10 Hz dSC stimulation evoked minimal motor effects (e.g. head turning), avoiding gross motor effects that confound the interpretation of plethysmographic recordings. Trials were conducted with rats in quiet wakefulness with at least 5 min between stimuli.

**Tail thermography.** During the light phase of the test day rats were individually transported to the laboratory (ambient temperature 23°C), nesting removed, and the home box placed within a high-walled black wooden box with an infrared camera 1 m overhead. Rats were allowed to habituate in low lighting for 15 min before fibre ferrule cleaning, lubrication and attachment to a rotary-coupled patch cable, followed by further habituation for at least 15 min or until tail temperatures stabilized as indicated by thermography. Video recording and stimulation were initiated using an automated digital sequence to record 10 min baseline behaviour, 10 min 1 Hz dSC stimulation (320 mW/mm², 20 ms pulse width) and 10 min post-stimulation activity.

### Terminal electrophysiology experiments

**Physiological effects of dSC stimulation.** Rats were anaesthetized with urethane, placed on a thermostatically controlled heating pad (Harvard apparatus, MA, USA: core temperature 36.5–37.5°C), the flank shaved, and the carotid artery and jugular vein cannulated to enable monitoring of arterial blood pressure (AP) and provide intravenous access, respectively. The trachea was intubated with a 14 g cannula, and the rat placed in a stereotaxic frame. Anaesthetic depth was monitored throughout surgery via assessing respiratory, pressor and motor responses to firm hind paw or tail pinch. Supplementary urethane (10% initial dose, i.v.) was administered as required.

The left postganglionic splanchnic sympathetic nerve was isolated via a retroperitoneal approach, mounted on bipolar silver electrodes and embedded in silicon polymer (Silgel,Wacker Chemie AG). Central respiratory activity was monitored via direct recording of the phrenic nerve or diaphragmatic electromyogram (EMG), recorded via Teflon-coated steel wire electrodes (Le et al., 2016). Signals were amplified, bandpass filtered (100–1000 Hz, BMA-400, CWE Inc.) and sampled at 3000–5000 samples/s (Spike 2). In most cases experiments were conducted in spontaneously breathing animals, supplemented with oxygen; in 4/19 experiments rats were vagotomized, subject to neuromuscular blockade with pancuronium bromide and artificially ventilated.

In experiments that measured the physiological effects of dSC-GiA terminal activation, rats were placed in a nose-down position and the dorsal medulla exposed. The caudal pole of the left facial nucleus was electrophysiologically mapped using field potentials antidromically evoked by facial nerve simulation, measured using a 1 MΩ glass pipette containing 3 M KCl. A 200 μm optical fibre was positioned 0.5–0.75 mm lateral to midline at rostrocaudal coordinates corresponding to the caudal margin of the facial nucleus, at a depth 300 μm

ventral to the position at which the largest facial field potentials were recorded, which reliably coincided with the GiA region. Fibre positions were reconstructed *post hoc* from tracks visible in serial medullary sections.

**GiA neuronal responsiveness to dSC stimulation.** Rats were prepared as described above, bilaterally vagotomized and artificially ventilated in oxygen-enriched room air under neuromuscular blockade. The T2 spinal cord was exposed and a bipolar concentric stimulating electrode (Rhodes NE-100) aimed at the dorsolateral funiculus (to antidromically activate the axons of bulbospinal neurons) at coordinates that resulted in maximal sympathetic nerve activity (SNA) responses to 200 $\mu$A cathodal pulses (0.2 ms). The facial nucleus was mapped as described above and extracellular recordings of spontaneously active neurons made via glass microelectrodes filled with 0.9% NaCl (30–50 M$\Omega$), amplified and filtered (Axoclamp 900A, Molecular Devices, Palo Alto, CA, 300–3000 Hz) and sampled at 10,000 samples/s using Spike 2. The region corresponding to the GiA (0.4–0.9 mm from midline, 2.8–4.1 mm deep to the dorsal surface, ±0.4 mm rostrocaudal to the caudal pole of the facial nucleus) was surveyed. Spontaneously active neurons with spinal projections were identified by constant-latency antidromic responses to spinal stimulation that collided with spontaneous orthodromic spikes (Lipski, 1981). Putative sympathetic premotor neurons were functionally identified from systole-triggered averages of neuronal discharge or spike-triggered averages of SNA, defining features of other sympathetic premotor populations (Byrum et al., 1984; Chen & Toney, 2010; Kanbar et al., 2011; McMullan et al., 2007; Morrison et al., 1988).

### Killing, perfusion, histology and imaging

At the conclusion of urethane-anaesthetized *in vivo* experiments, the animals were overdosed with 150 mg kg$^{-1}$ sodium pentobarbital in saline delivered I.V., then perfused with 500 ml heparinized saline followed by 500 ml 4% paraformaldehyde. Conscious animals, at the conclusion of experiments, were overdosed with sodium pentobarbital I.P. (200 mg kg$^{-1}$ in 0.9% saline (1:4 dilution)), then perfused with 500 ml heparinized saline followed by 500 ml 4% paraformaldehyde. Brains were removed, post-fixed at room temperature overnight, and cut into 50 $\mu$m thick coronal sections using a vibrating microtome (Vt1200S, Leica) at 200 $\mu$m intervals.

To verify dSC injection sites, sections were mounted immediately onto glass slides with mounting medium (Dako Fluroshield with DAPI, Sigma), cover slipped and native reporter fluorescence imaged using a Zeiss Z2 epifluorescence microscope at 10×. Pilot experiments indicated weak or absent behavioural responses from

off-target injections: 16 successful experiments were chosen to generate a heat map demarcating reporter expression in experiments associated with typical behavioural responses. To construct the heat map, three greyscale images were used from each animal: one that contained the centre of the injection site, one 200 $\mu$m rostral to the centre and one 200 $\mu$m caudal to the centre. Images were downsampled to 25%, Gaussian blurred (6 pixel) and eYFP fluorescence normalized so that the value of the brightest labelling became white and the background became black. Images were aligned to the appropriate rat atlas plates (Paxinos & Watson, 2006) and individual images from each bregma level overlaid and averaged using ImageJ. Normalized averaged greyscale images were then colourized using the ImageJ Thermal LUT.

'On target' experiments were defined as having reporter expression distributed 6.8–7.8 mm caudal to the bregma, mostly within the deep grey and deep white layers of the SC, medial to the nucleus of the brachium of the inferior colliculus, dorsal to the precuniform area, and lateral to the dorsolateral and lateral periaqueductal grey. If YFP reporter expression or optical fibre tracks fell outside these boundaries, the data were excluded from the analysis.

**Immunohistochemistry.** Brain sections were permeabilized for 3 × 15 min in TPBS with 0.1% Tween20 or 0.2% Triton-100 and blocked for non-specific binding in TPBS containing 2% bovine serum albumin and 0.2% Triton-100 for 1 h at room temperature. Primary antibodies were added to blocking buffer and incubated for 48 h at 4°C. Sections were then washed in PBS 3 × 30 min and incubated in TPBSM with 5% NHS and secondary antibodies for 12 h at 4°C. Sections were washed again for 3 × 20 min in PBS before being mounted on glass slides with Dako Fluroshield mounting medium with DAPI and cover slipped.

Fluorescence imaging was performed on a Zeiss Z2 epifluorescence microscope or Leica TCS SP5X confocal microscope. Images were prepared and analysed using ImageJ Fiji and Corel Photopaint X7.

### Statistical analysis

Data are reported as means ± standard deviation unless otherwise indicated. Graphpad Prism 7 was used for all statistical analysis.

**Effects of cl-dSC stimulation on awake rats.** The percentage of time spent in the centre of the OFT arena, and the percentage of time spent in open, closed or end zones of the EPM, were compared between ChR2 and control animals by unpaired *t* test. The numbers of low- and high-frequency USVs were also compared between

ChR2 and control groups using an unpaired *t* test. The respiratory effects of cl-dSC stimulation were quantified by comparing the average respiratory frequency in the 10 s before stimulation with the maximal respiratory frequency recorded during stimulation (running average, 3 s time constant). Significant differences in effect size between control and ChR2 animals were identified using the Mann–Whitney test. Fast Fourier Transformations of EEG activity recorded 10 s prior to and during cl-dSC stimulation were constructed in Spike 2 (Hanning window, 2048 Hz). EEG frequency bands were Delta: 0.5–4.5 Hz; Theta: 4.5–9 Hz; Alpha: 9–13 Hz; Beta: 13–20 Hz (Burke, Kanbar, Viar et al., 2015). EEG frequency bands were expressed relative to total power between 0 and 20 Hz, and compared using repeated measures (RM) two-way ANOVA followed by Sidak's *post hoc* test where indicated by significant interaction.

Tail temperature was quantified from infrared video images sampled at 10 s intervals. For each image a region of interest was manually drawn around the tail and temperature estimated from the greyscale intensity of the brightest (i.e. warmest) pixel (black = 20°C, white = 36°C) using ImageJ Fiji. Each data point was expressed relative to the average temperature recorded during the baseline period for that animal and smoothed (five-point rolling average). Significant differences in the effects of photostimulation on tail temperature in ChR2 and control animals were identified using RM two-way ANOVA. Recordings with unstable tail temperature during the baseline period were excluded from the analysis.

The effects of intermittent stimulation of the left dSC on motor behaviour was extracted from the same videos: the number of complete rightward circles made during stimulation was counted; significant differences in circling behaviour between ChR2 and control animals were detected using Student's *t* test. The distance travelled before, during and after stimulation was measured using Tracker software (Open Source Physics): videos were downsampled to one frame/second and the position of the nose manually marked. The distance travelled from frame to frame was smoothed (nine-frame rolling average) and the effects of time and dSC stimulation determined by RM two-way ANOVA.

**Effects of cl-dSC stimulation on anaesthetized rats.** To assess the effects of 10 or 20 Hz stimulation of the dSC or dSC-GiA terminals, systolic AP and rectified SNA were smoothed (1 s time constant), downsampled to 10 Hz and exported in contiguous blocks that started 20 s prior to and ended 30 s after the onset of stimulation; changes in systolic AP were expressed relative to the average baseline value; SNA was normalized relative to baseline (100%) and noise (0%: obtained by hexamethonium bromide at the conclusion

of experiments: 5 mg I.V. ($\sim$10 mg kg$^{-1}$; Sigma Aldrich, Australia), as previously described (Underwood et al., 2022). Significant differences between responses to cl-dSC stimulation in ChR2 and control animals, or between stimulation of cl-dSC cell bodies or their GiA terminals, were identified by RM two-way ANOVA. Heart and respiratory rates were derived from systolic AP and the leading edge of diaphragmatic EMG/phrenic neurogram burst, respectively, and were quantified by measuring the maximum deviation of the smoothed running averages (bin width 1 and 3 s, respectively) from baseline. Statistically significant differences in responsiveness of ChR2 and control animals were detected using an unpaired *t* test. Ensemble averages of rectified smoothed ($\tau = 20$ ms) SNA responses to dSC or dSC-GiA terminal stimulation were compiled from 200–600 consecutive stimuli delivered at 0.5 Hz.

To assess the effects of cl-dSC stimulation on GiA activity, the neuronal discharge was plotted as mean frequency (2 s time constant) and the effects of optical cl-dSC stimulation at 10 or 20 Hz expressed as maximum change in discharge during stimulation compared with the average discharge in the 10 preceding seconds. Responses to low-frequency cl-dSC stimulation were identified from peristimulus time histograms constructed from 200–400 consecutive trials. Spike-triggered averages of rectified and smoothed (50 ms time constant) SNA were generated from 400–26,000 sweeps triggered by the spontaneous discharge of GiA neurons. 'Dummy' spikes that occurred at the same average frequency were used as a negative control (Morrison et al., 1988; Turner et al., 2013).

## Results

### Behavioural effects of optogenetic stimulation of the dSC in awake rats

Following viral transduction of the cl-dSC to induce Channelrhodopsin2-YFP (ChR2, Fig. 1*A*) or reporter (control) expression, we first compared the effects of brief optogenetic stimulation (63–160 mW/mm$^2$, 5–10 Hz, 20 ms pulses, 5–10 s) on behaviour in awake rats. In confirmation of previous findings in mice (Isa et al., 2020), brief stimulation of the cl-dSC evoked orienting behaviour, first detectable as a pause in ongoing activity with contraversive head turning (i.e. turning in the direction contralateral to the side stimulated) in ChR2 rats. This response was reproducible within and between animals and was not seen in animals injected with a control vector or in animals in which ChR2 reporter expression fell outside of the cl-dSC. As stimulus intensities increased, head-turning became more pronounced and led to transient full-body turning on the spot (Fig. 1*B*). At the highest stimulus intensities tested

(320 mW/mm$^2$, 10–20 Hz, 20 ms pulses, 5–10 s), the initial pause and head-turning component of the response did not occur, and instead rats exhibited full-body turning on the spot, as reported by Isa et al. (2020) in the mouse. Coordinated locomotor responses, such as running, were rare and clear defensive behaviours, such as freezing, explosive escape or retreat, typical of stimulation of the rostromedial tectal defence pathway (Isa et al., 2020), were only seen in 1/33 rats. Orienting towards the ipsilateral side was never seen. Immediately following cl-dSC stimulation, the motor effects abruptly ceased and rats

reverted to their previous behaviours without discernible signs of fear or heightened vigilance, such as aversion to approach or handling.

To test the behavioural effects of more sustained stimulation of the cl-dSC, we adapted the stimulus protocol (320 mW/mm$^2$, 1 Hz, 20 ms pulses, 1–2 min), after which rats were immediately transferred onto either an open field or an EPM apparatus. These stimulation parameters continued to evoke orienting and motor responses, quantified as an increase in circling (Fig. 1*C1*) and locomotion (measured by overhead video tracking of

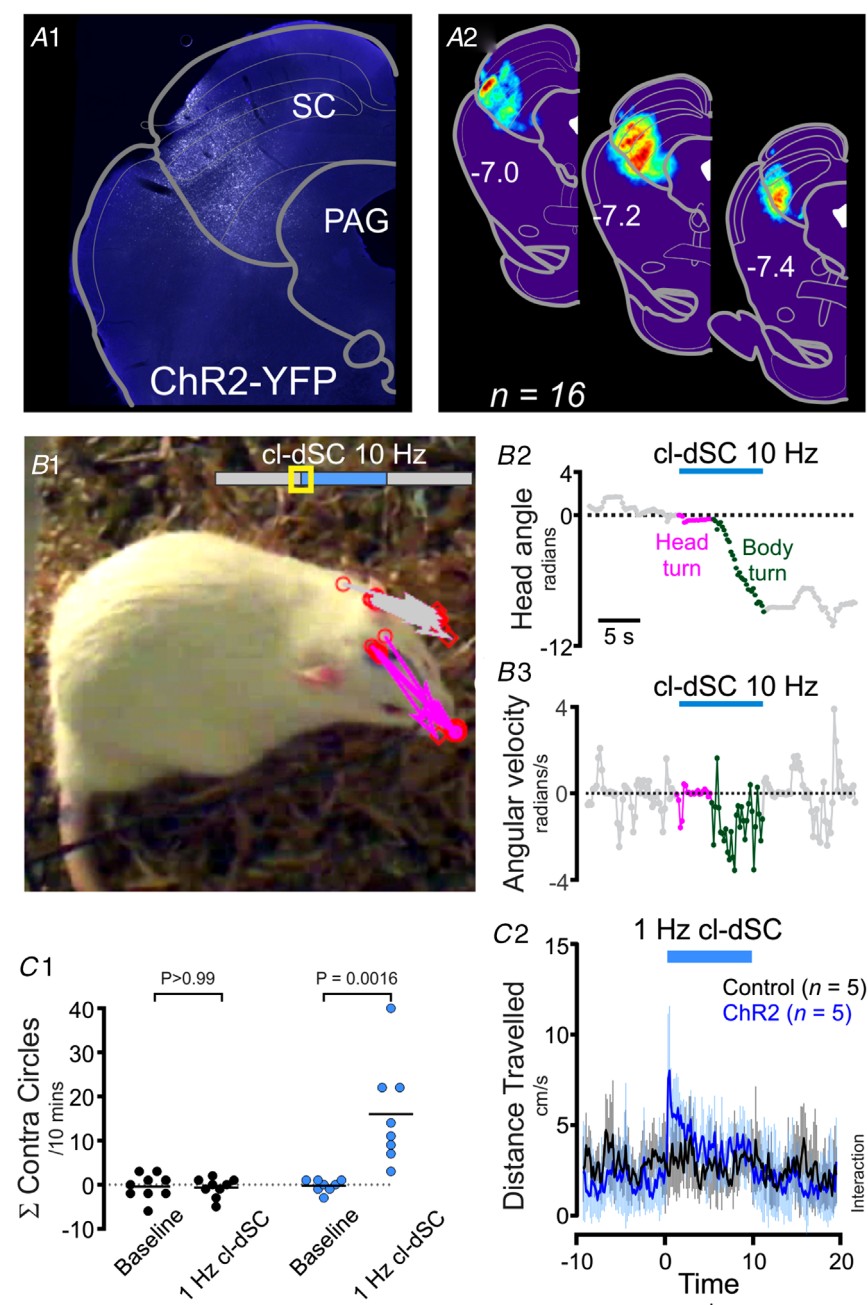

**Figure 1. Optogenetic dSC stimulation causes orienting and circling behaviours in awake rats**
*A1*, distribution of ChR2-YFP fluorescence in a representative experiment and (*A2*) normalized ChR2-YFP distribution from 16 animals. Optogenetic stimulation of this region evoked contraversive orienting: Panel *B* depicts movements evoked by moderate-intensity cl-dSC stimulation in a typical experiment. *B1*, an annotated video frame captured after 1 s of stimulation; head position in preceding frames is indicated by arrows drawn between the optical fibre and tip of the nose in grey (baseline) and magenta (stimulation). *B2* and *3*, head angle and angular velocity during the same experiment. *C*, behavioural effects of sustained cl-dSC stimulation in rats injected with ChR2 and control vectors. Incidence of contraversive circling (*C1*) and distanced travelled (*C2*) during 10 min 1 Hz cl-dSC stimulation in ChR2-treated and control rats. **$P < 0.01$ paired *t* test, ####interaction $P < 0.0001$, two-way repeated measures ANOVA. Error bars indicate standard deviation from the mean throughout. [Colour figure can be viewed at wileyonlinelibrary.com]

the nose: Fig. 1*C2*), particularly during the first 2 min of stimulation.

On the other hand, prior cl-dSC stimulation was not associated with detectable changes in open field exploration (*n* = 9 ChR2 *vs*. 9 control rats, unpaired *t* test, *P* = 0.621) or plus maze exploration (*n* = 10 ChR2 *vs*. 9 control rats, unpaired *t* test, *P* > 0.99, Fig. 2*A*). Similarly, activation of the cl-dSC was not associated with changes in the number of low- or high-frequency USVs, indices of aversive and appetitive stimuli, respectively (Brudzynski, 2007, 2013), during stimulation in the home cage (unpaired *t* test, *P* = 0.218, *P* = 0.925, respectively, *n* = 6 ChR2 *vs*. 5 control rats, Fig. 2*B*). Thus in summary, optogenetic cl-dSC stimulation evoked immediate, transient and stereotypic orienting-like motor responses that were not associated with anxiety-like behaviours.

### Effects of optogenetic dSC stimulation on EEG, respiratory rate and tail vasoconstriction in awake rats

The orienting responses evoked by cl-dSC stimulation were accompanied by profound transient elevations in respiratory rate (Fig. 3*A*), from 2.16 ± 1.0 Hz (baseline) to a maximum of 4.62 ± 1.57 Hz with photostimulation at 10 Hz (Fig. 3*A2*). Stimulation of the cl-dSC also produced EEG desynchronization in ChR2 rats (reduced EEG delta power: 36 ± 19% (baseline) *vs*. 18 ± 7% (ChR2-stim); *n* = 6, *P* = 0.031, Sidak) together with increased theta power(43 ± 16% (baseline) *vs*. 62 ± 11% (ChR2-stim); *n* = 6, *P* = 0.022, Fig. 3*A1* and *A3–5*), even at stimulation intensities that evoked minimal motor effects (63–160 mW/mm², 5–10 Hz, 20 ms pulses, 10 s, head-turning only). No effect on EEG power spectrum was observed in control animals (Fig. 3*AC*; see the Statistical Summary table for data on EEG power bands in control animals). In the rat, orienting responses to innocuous naturalistic stimuli are associated with transient sympathetically mediated tail

vasoconstriction (de Menezes et al., 2009; Nalivaiko et al., 2012), resulting in tail-cooling and core heat retention. Infrared tail thermography revealed similar effects in response to cl-dSC stimulation (320 mW/mm², 1 Hz, 20 ms pulses, 10 min): tail temperature dropped to a nadir of −2.0 ± 1.4°C during photostimulation and then rebounded to a peak of +1.1 ± 1.0°C relative to baseline over the 10 min after stimulation (RM two-way ANOVA interaction *P* < 0.0001, Fig. 3*B*). Activation of respiratory and autonomic outputs is therefore an inherent component of orienting responses evoked by cl-dSC stimulation.

### Autonomic and respiratory effects of optogenetic dSC stimulation under anaesthesia

An increase in somatomotor activity activates skeletal muscle receptors, leading to reflex increases in sympathetic and respiratory activities (Murphy et al., 2011; Shevtsova et al., 2019). To differentiate autonomic and respiratory responses directly evoked by cl-dSC stimulation from those that may be secondary to motor components of orienting responses, recordings of respiratory, blood pressure, heart rate and splanchnic sympathetic nerve responses to cl-dSC stimulation were made in urethane-anaesthetized rats (Fig. 4). Low-frequency stimulation (320 mW/mm², 0.5 Hz, 20 ms pulses, 200–600 repeats) revealed constant-latency sympathoexcitatory responses in 8/11 cases (onset 43 ± 8 ms, peak 101 ± 8 ms, Fig. 4*D*). In experiments where sympathetic responses were observed, intensity-dependent respiratory and cardiovascular responses to short trains of repetitive cl-dSC stimulation were also observed (320 mW/mm², 10–20 Hz, 20 ms pulses, 10 s). Responses were characterized by abrupt increases in respiratory frequency and respiratory burst size (Fig. 4*C*, 8/8 cases), increased mean splanchnic SNA (Fig. 4*E*, 6/8 cases), and increased systolic arterial blood pressure (sAP, Fig. 4*F*, 7/8 cases), with variable effects on heart rate (Fig. 4*G*).

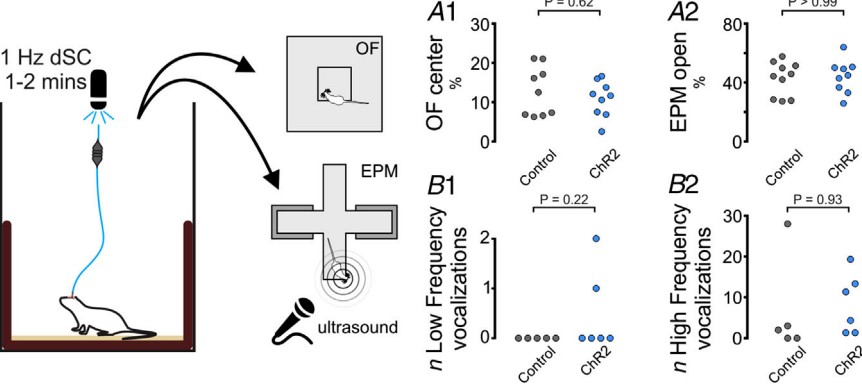

**Figure 2. Optogenetic dSC stimulation is not associated with anxiety-like behaviours or ultrasonic vocalisation**
The presence of anxiety-like behaviours was assayed immediately after optical cl-dSC stimulation using the open field (OF, *A1*) or elevated plus maze (EPM, *A2*) apparatus: no significant difference was detected between rats treated with ChR2 or control vectors. Similarly, no effects on the incidence of low- or high-frequency ultrasonic vocalizations were detected while exploring the EPM (*B1* and *B2*). [Colour figure can be viewed at wileyonlinelibrary.com]

## Pontomedullary projections of cl-dSC neurons

The short latencies of splanchnic sympathetic responses to cl-dSC stimulation, and our previous observation that cl-dSC-facilitated autonomic responses to sensory stimulation persist following pre-collicular decerebration (Muller-Ribeiro et al., 2014), suggest a direct descending neural pathway from the cl-dSC that does not involve ascending loops through forebrain regions. Using anterograde tract tracing, we identified putative connections between cl-dSC output neurons and hindbrain autonomic and respiratory relays. Microinjection of AAV-CBA-tdTomato into the cl-dSC revealed a clearly defined ascending projection to the thalamus and hypothalamus, a projection to the contralateral superior and inferior colliculi via the commissural nucleus of the inferior colliculus, and a descending projection to the brainstem (Fig. 5*A*). Confocal imaging of tdTomato-labelled hindbrain terminal fields identified close appositions in candidate autonomic nuclei (Fig. 5*B*), including moderate innervation of catecholaminergic neurons in the A5 (Fig. 5*C*), A6 (locus coeruleus, Fig. 5*D*) and A7 (parabrachial/Kölliker-Fuse, not shown) regions.

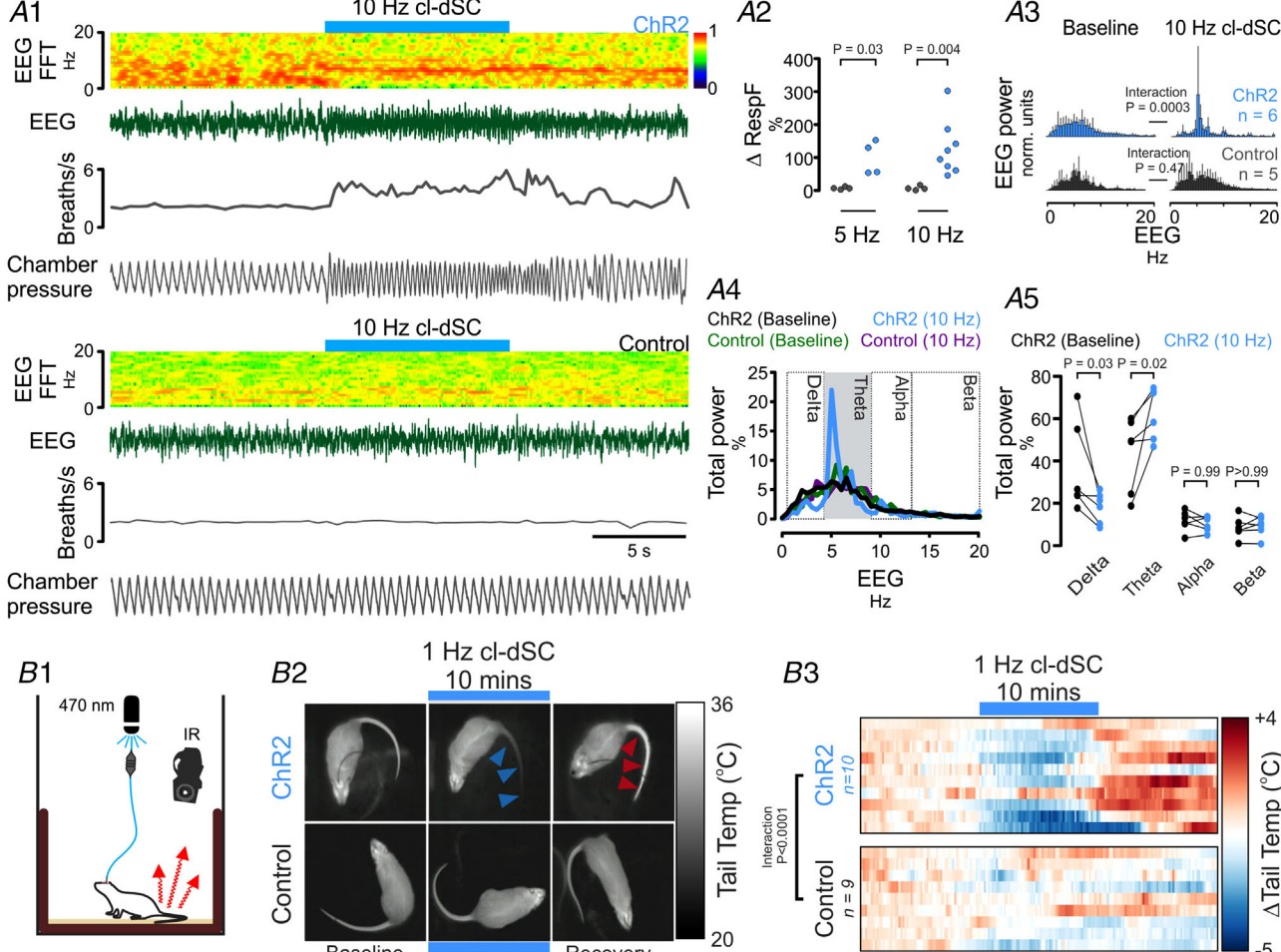

**Figure 3. cl-dSC stimulation drives arousal, ventilation and tail vasoconstriction in ChR2-treated but not vector control rats**

*A1*, simultaneous recording of electroencephalogram (EEG) and respiratory rate from ChR2 (upper) and control vector treated (lower panels) animals showing tachypnoea and EEG desynchronization (0.5–4.5 Hz) with increased theta power (4.5–9 Hz). Pooled data shown in *A2* (increase in respiratory frequency; *,**Mann-Whitney $P < 0.05$, 0.01), *A3*, pooled EEG (0.5–20 Hz) before and during photostimulation in ChR2 and control rats; frequency bands are defined in *A4*. *A5*, changes in Delta and Theta power in ChR2 rats as a percentage of total EEG power, *$P < 0.05$ (Sidak's *post hoc* test after two-way repeated measures ANOVA). *B1*, tail temperature was measured by infrared thermography before, during and after 10 min of 1 Hz cl-dSC stimulation. *B2*, thermal imagery illustrates tail-cooling in ChR2 experiment during stimulation (blue arrowheads), and tail temperature rebound during the recovery period in a representative experiment (red arrowheads). *B3*, heat maps illustrate recordings from 10 ChR2 and nine control vector treated rats: each row represents an individual experiment ###, #### two-way repeated measures ANOVA interaction $P < 0.001$, $<0.0001$. [Colour figure can be viewed at wileyonlinelibrary.com]

A more conspicuous and largely ipsilateral nexus of fibres and arborized varicosities extended across subnuclei within the ventromedial medulla, most densely within the gigantocellular reticular nucleus alpha part (GiA) dorsal to the pyramidal tract but extending also into surrounding nuclei (lateral paragigantocellular reticular nucleus and gigantocellular reticular nucleus ventral part) at the level of the caudal pole of the facial nucleus (Fig. 5*E*). There was also light labelling in the obscurus, pallidus and magnus raphé nuclei, although putative terminals were not identified on serotonergic tryptophan hydroxylase-immunoreactive neurons (data not shown).

## Connections between descending projections from the cl-dSC and spinally projecting neurons in the ventromedial medulla

Spinally projecting neurons in this region of the ventromedial medulla, including V2a neurons that express the

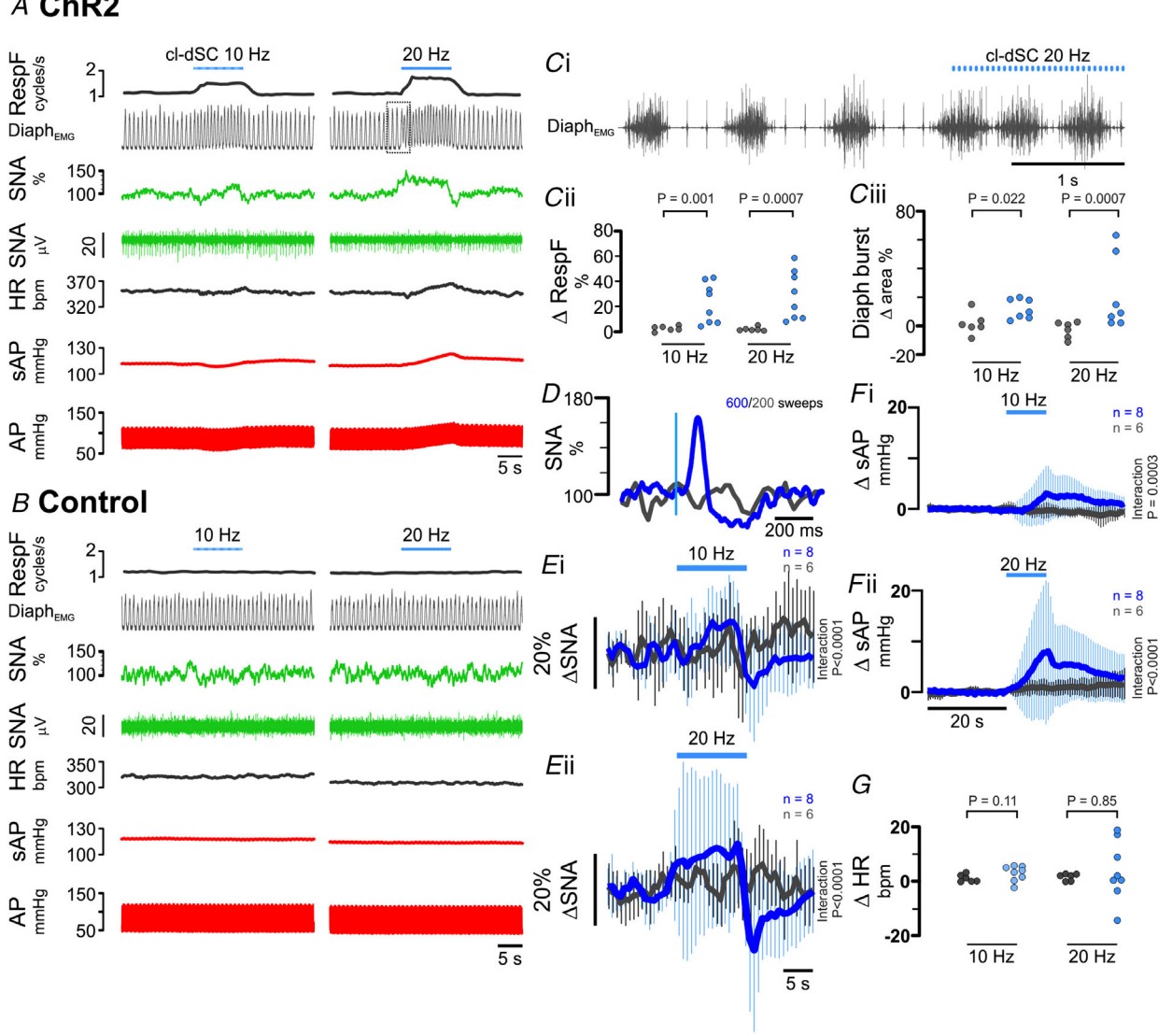

**Figure 4. Respiratory and cardiovascular effects of cl-dSC stimulation under urethane anaesthesia**
Representative recordings from rats treated with ChR2 (*A*) and control (*B*) vectors. Boxed region in Panel *A* denotes expanded trace shown in Panel *Ci*. Respiratory effects (Panel *C*) manifested as an increase in diaphragmatic EMG burst frequency (*Cii*) and amplitude (*Ciii*). Single-pulse cl-dSC stimulation evoked short-latency monophasic bursts in SNA (*D*: vertical blue line denotes laser onset); repetitive cl-dSC stimulation evoked frequency-dependent recruitment of SNA (*E*) and systolic blood pressure (sAP, *F*), with variable effects on heart rate (HR, *G*). Bold traces in *E* and *F* indicate mean; light traces indicate SEM. *,**,***Mann-Whitney $P < 0.05$, $<0.01$, $<0.001$; ####two-way repeated measures ANOVA interaction $P < 0.0001$. [Colour figure can be viewed at wileyonlinelibrary.com]

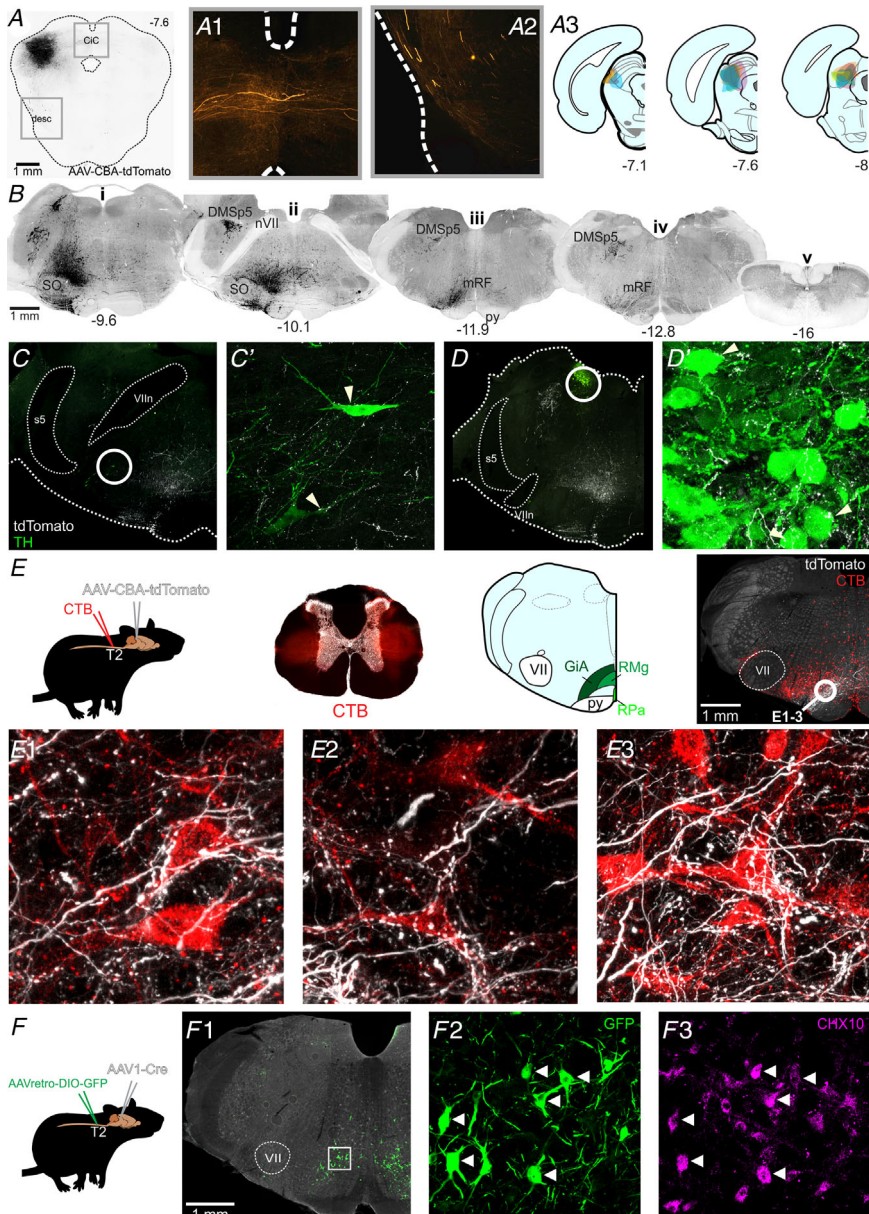

**Figure 5. Anterograde labelling from the cl-dSC**

*A*, example of AAV-CBA-tdTomato injection site in the cl-dSC; fibres emerging from the injection site projected to the contralateral colliculi via the commissure of the inferior colliculus (*CiC*, *A1*) or followed a descending trajectory via the lateral midbrain (desc, *A2*). *A3*, composite diagram showing distribution of injection sites in six experiments. *B*, histological sections from one experiment, illustrating the brainstem distribution of tdTomato labelling. The descending bundle split into two tracts in the pons, a more substantial ventral bundle that surrounded the superior olive on all sides (SO, *Bi*), and a lesser and exclusively ipsilateral dorsal tract that innervated the dorsomedial spinal trigeminal nucleus (DMSp5, *Bii*) with large-calibre fibres. The ventral branch occupied a column extending from the ventral surface to an apex at the locus coeruleus, was sparsely mirrored on the contralateral brainstem, and comprised both large- and fine-calibre fibres and putative terminals. Caudal to the facial nerve (nVII, *Bii*) the ventral branch was most densely concentrated in the medullary reticular formation (mRF, *Biii*) dorsal to the pyramidal tract (py) at the level of the caudal pole of the facial nucleus. Labelled fibres were infrequently encountered in the caudal medulla (*Biv*) and virtually absent from spinal sections (*Bv*). Coordinates indicate the rostrocaudal position with respect to the bregma. Putative targets of the cl-dSC outputs included rare tyrosine hydroxylase (TH) immunoreactive neurons in the A5 (*C*) and A6 (*D*) cell groups, and a conspicuous terminal field in the gigantocellular alpha region of the reticular formation (GiA, *E*). Putative cl-dSC projections to spinally projecting neurons in the GiA were identified in experiments in which anterograde cl-dSC labelling was combined with retrograde labelling of spinally projecting neurons by microinjection of cholera toxin B (CTB) at the thoracic intermediolateral cell column: top row of panel *E* shows a schematic of the

experimental strategy; brightfield transverse thoracic spinal cord section superimposed with CTB injection sites in red; and an atlas plate highlighting the GiA, midline raphe magnus (RMg) and pallidus regions (RPa) with a low-power image showing the distribution of CTB-labelled cell bodies (red) and cl-dSC terminals (white). *E1–3*, high-power confocal images from the circled region show close apposition between cl-dSC terminals and spinally projecting GiA neurons. *F*, similar results were obtained using AAV1-mediated trans-synaptic tagging: spinally projecting neurons were transduced by injection of Cre-dependent AAVretro-hSyn-DIO-GFP in the thoracic spinal cord, GFP expression controlled by the trans-synaptic trafficking of AAV1-hSyn-Cre from cl-dSC. *F1*, GFP-immunoreactive neurons were exclusively found in GiA and included CHX10-immunoreactive V2A neurons (arrowheads) and CHX-10 negative cells (*F2* and *3*). [Colour figure can be viewed at wileyonlinelibrary.com]

transcription factor CHX10, are considered key players in the initiation and arrest of locomotion and orienting behaviours (Dougherty & Kiehn, 2010; Usseglio et al., 2020). We confirmed the presence of tdTomato-labelled cl-dSC terminals in close apposition to putative bulbo-spinal neurons, identified by retrograde transport of cholera toxin B (CTB) from the thoracic spinal cord (Fig. 5*E1–3*), and to putative V2a neurons, identified by CHX10 immunoreactivity (not shown), with the caveat that not all close appositions identified under light microscopy are functional synapses (reviewed by Saleeba et al., 2019). To overcome this limitation, we used an intersectional viral tracing strategy, based on the anterograde trans-synaptic trafficking of AAV1 vectors (Zingg et al., 2017), to selectively label spinally projecting neurons that receive synaptic input from the cl-dSC. In these experiments AAV1-hSyn-Cre was injected into the cl-dSC and a Cre-dependent retrograde vector, AAVretro-hSyn-DIO-GFP, at the intermediolateral column of the T2 spinal cord (Fig. 5*F*), such that only spinally projecting neurons that are post-synaptic targets of cl-dSC neurons contain both vectors, permitting GFP transcription. This approach reproduced the findings of our conventional anterograde tracing experiments, revealing GFP-immunoreactive neurons concentrated in the GiA in three of four rats, of which 38% were found to be CHX10 immunoreactive (14/38 neurons, four confocal stacks from one animal, Fig. 5*F2–3*). No GFP+ neurons were identified in any other region except for a small number of neurons in the sub-coeruleus region of the pontine reticular nucleus ($n$ = 1/4 rats). These findings therefore provide evidence for a disynaptic pathway that links the cl-dSC to thoracic spinal outputs via a relay in the GiA.

The lateral aspect of the medullary terminal field encroached upon the medial border of the rostral ventro-lateral medulla (RVLM) C1 region, and under high magnification, sparse fine fibres and varicosities could be observed throughout its medial third. Given the key role played by the RVLM as a sympathoexcitatory relay, we closely examined C1 neurons for evidence of collicular terminal labelling. In contrast to the rostral ventro-medial medulla, no evidence of terminal appositions on catecholaminergic TH-immunoreactive RVLM neurons or spinally projecting CTB-immunoreactive RVLM neurons was identified (data not shown). We also examined anatomical regions that house components of the central respiratory pattern generator for potential terminal labelling. We found no evidence of terminals immediately ventral to nucleus ambiguus in the region immediately caudal to the facial nucleus, corresponding to the region that contains most Bötzinger neurons (Le et al., 2016). Although fine fibres were occasionally encountered in more caudal sections corresponding to the region of the pre-Bötzinger complex, no evidence of appositions between putative axonal varicosities and neurons immunoreactive for the neurokinin-1 receptor (NK1R), considered a marker for inspiratory inter-neurons, were observed (data not shown).

### Autonomic and respiratory responses to cl-dSC stimulation are mediated via a relay in the ventromedial medulla

Neurons in the ventromedial medulla, including the GiA, form monosynaptic connections with spinal motoneurons (Esposito et al., 2014) and have recently been implicated in coordinating the somatomotor effects of dSC stimulation (Isa et al., 2020). However, the GiA region is also a major source of sympathetic premotor neurons that innervate vasomotor and non-vasomotor targets (Aicher et al., 1995; Babic & Ciriello, 2004; Jansen et al., 1995; Kerman, 2008; Strack et al., 1989). In addition, V2a neurons within the GiA have been implicated in regulating respiratory function (Crone et al., 2012). We therefore tested the possibility that the disynaptic pathway from the cl-dSC to the spinal cord may mediate autonomic and respiratory responses evoked from the cl-dSC in addition to pre-viously established motor functions.

For this purpose, we injected AAV-Syn-ChR2-YFP into the cl-dSC and optically stimulated ChR2-transduced axon terminals in the GiA in urethane-anaesthetized rats. Autonomic responses were consistent with those observed following cl-dSC stimulation, and were only observed in rats in which expression of ChR2-YFP in the ipsilateral GiA was histologically verified (Fig. 6). In 4/5 animals tested, low-frequency GiA stimulation (320 mW/mm², 0.5 Hz, 20 ms pulses) evoked sympathoexcitatory responses that occurred at a shorter latency (onset $30 \pm 2$ *vs.* $43 \pm 8$ ms, unpaired $t$ test $P$ = 0.021) but equivalent

peak amplitude (151 ± 28 *vs.* 163 ± 70%, unpaired *t* test *P* = 0.76) compared with cl-dSC stimulation (Fig. 6*E*).

The profiles of splanchnic sympathetic and pressor responses to 20 Hz GiA stimulation were not significantly different from those evoked by 20 Hz cl-dSC stimulation (two-way ANOVA interaction *P* > 0.99 for both sympathetic and pressor responses, *n* = 7, direct comparison in Fig. 7*A*). On the other hand, although GiA

stimulation increased respiratory frequency in 4/7 cases, the effects were consistently smaller than those evoked by cl-dSC stimulation (10 ± 8 *vs.* 29 ± 19%, unpaired *t* test *P* = 0.036, Fig. 7*B*).

The results of these experiments are consistent with a critical role for the cl-dSC-GiA pathway in mediating the autonomic components of responses to cl-dSC stimulation. However, optogenetic stimulation of axons

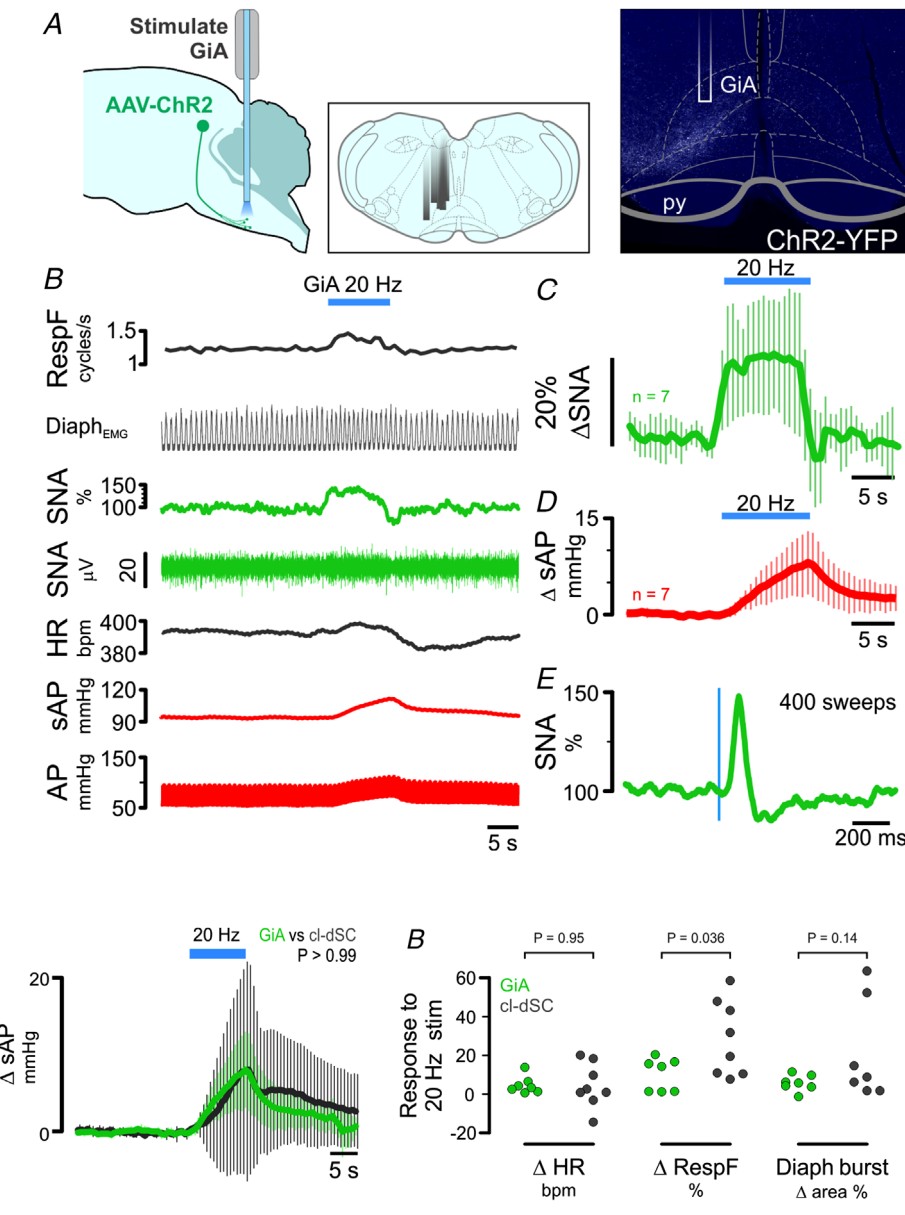

**Figure 6. Respiratory and cardiovascular responses to optogenetic stimulation of cl-dSC-GiA terminals under urethane anaesthesia**
*A*, illustration of (L–R) experimental strategy, locations of optical fibres in different experiments and a histological specimen illustrating fibre position and terminal ChR2 labelling. *B*, physiological recording of respiratory, sympathetic and cardiovascular responses to GiA stimulation; pooled sympathetic and pressor responses from seven experiments are shown in *C* and *D*. *E*, stimulus-triggered average of sympathetic response to 0.5 Hz stimulation from the experiment shown in *B*. [Colour figure can be viewed at wileyonlinelibrary.com]

**Figure 7. Direct comparison of responses evoked by optogenetic stimulation of cl-dSC neurons (dark grey) and their GiA terminals (green)**
Outputs examined were sympathetic nerve activity and systolic blood pressure (*A*), and heart rate, respiratory frequency and respiratory burst area (*B*). Significantly different responses were only detected in respiratory frequency (\**P* < 0.05, unpaired *t* test). [Colour figure can be viewed at wileyonlinelibrary.com]

and terminals can also drive antidromic activation of cell bodies and axon collaterals (Sato et al., 2014), and so we sought independent evidence that putative GiA sympathetic premotor neurons are directly activated by cl-dSC stimulation. In two urethane-anaesthetized rats, extracellular recordings were made from seven spontaneously active (range 1–39 Hz, median 7 Hz) spinally projecting GiA neurons (conduction velocity 12.5 ± 7.8 m/s, Fig. 8*A* and *B*). All seven neurons exhibited frequency-dependent excitatory responses to cl-dSC optogenetic stimulation (320 mW/mm$^2$, 20 ms pulses, 10 Hz: range +8 to +64%, median +12%; 20 Hz: range +16 to +127%, median +73%, Fig. 8*C*). In two out of four cells tested, short latency (8 and 17 ms) excitatory responses were evoked by low-frequency cl-dSC stimulation (320 mW/mm$^2$, 0.5 Hz, 20 ms pulses, Fig. 8*D*). Four of the seven cells were identified as putative sympathetic premotor neurons based on the entrainment of unit discharge to blood pressure pulsatility ($n = 3$, Fig. 8*E*) or covariation of neuronal discharge with SNA, revealed by spike-triggered averaging ($n = 4$, data not shown).

## Discussion

The main new findings of this study are as follows. In conscious rats, optogenetic stimulation of the cl-dSC at intensities that drive orienting responses also evokes EEG desynchronization and theta rhythm, and respiratory and cutaneous vasomotor responses similar to those evoked by natural salient stimuli. Such responses occurred in the absence of detectable anxiety-like behaviours. In anaesthetized rats, optogenetic stimulation of the cl-dSC was also shown to evoke increases in blood pressure and splanchnic sympathetic activity. Anatomical experiments demonstrated monosynaptic connections of descending axons from cl-dSC neurons with V2A and non-V2A spinally projecting neurons in the GiA region in the ventromedial medulla. Optogenetic stimulation of cl-dSC efferent terminals in the GiA region mimicked the splanchnic and pressor responses, but not the respiratory response, evoked by direct stimulation of the cl-dSC. Spinally projecting GiA neurons, including those with functional properties consistent with sympathetic premotor neurons, are reliably activated by cl-dSC stimulation.

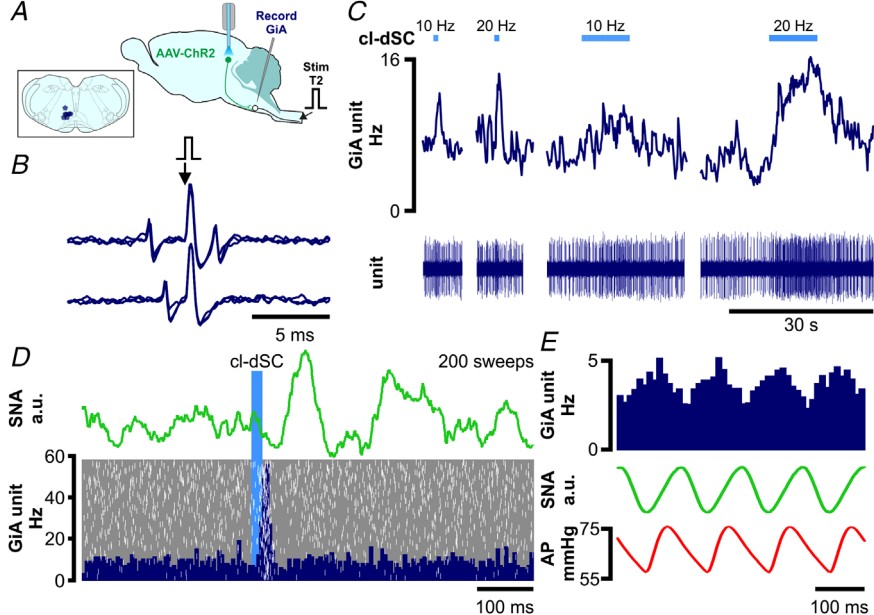

**Figure 8. Optogenetic cl-dSC stimulation activates spinally projecting cardiovascular and non-cardiovascular GiA neurons**
*A*, extracellular recordings were made from bulbospinal GiA neurons under urethane anaesthesia after transduction of cl-dSC neurons with ChR2 and fibre optic implantation; inset shows recording sites of seven neurons. *B*, spinally projecting neurons exhibited constant-latency antidromic spikes in response to electrical stimulation of the T2 spinal cord (top trace) which collided with spontaneous orthodromic spikes (lower trace). *C*, cl-dSC stimulation evoked frequency-dependent increases in baseline activity: trace shows typical responses to short (1 s) and long (10 s) trains of stimulation at 10 and 20 Hz (same neuron as *B*). *D*, low-frequency cl-dSC stimulation evoked short-latency response in the same neuron (laser-triggered peristimulus time histogram (dark blue) with overlaid raster (white spikes, grey background, lower panel) that preceded simultaneously recorded splanchnic sympathetic responses (green trace, upper panel)). *E*, putative sympathetic premotor neurons were identified by covariation of spontaneous neuronal activity (top trace) with SNA (green) and pulsatile arterial pressure (lower trace). [Colour figure can be viewed at wileyonlinelibrary.com]

## Pattern of autonomic and respiratory responses evoked from the cl-dSC

As described in the Introduction, novel salient alerting stimuli (which may be visual, auditory or somatosensory) typically evoke a response characterized by an increase in arterial pressure, cutaneous and mesenteric vasoconstriction, an increase in respiratory rate and EEG desynchronization (Baudrie et al., 1997; Caraffa-Braga et al., 1973; de Menezes et al., 2009; Kabir et al., 2010; Nalivaiko et al., 2012; Rettig et al., 1986; Yu & Blessing, 1997). Changes in heart rate, however, are generally small and variable (Baudrie et al., 1997; Kabir et al., 2010; Nalivaiko et al., 2012). In the present study, this same pattern of autonomic, respiratory and EEG changes was evoked by optogenetic stimulation of the cl-dSC. In conscious rats, these autonomic, respiratory and EEG responses consistently accompanied an orienting response, characterized by head-turning which in some cases was followed by full-body turning.

The question arises as to whether the autonomic and respiratory effects were a direct consequence of cl-dSC stimulation, or were secondary either to the behavioural somatomotor response or to the psychological effects evoked by the cl-dSC stimulation. Autonomic and respiratory responses were still evoked by cl-dSC optogenetic stimulation in urethane-anaesthetized rats with neuromuscular blockade, demonstrating that these responses were not dependent on somatomotor effects. In conscious rats, two observations indicated that the autonomic and respiratory responses were not a consequence of psychological stress either. First, the responses ceased abruptly on termination of the stimulus, in contrast to stress-evoked autonomic and respiratory responses which typically outlast stimulus presentation (Ootsuka et al., 2017; Vianna & Carrive, 2005; Vianna et al., 2008). Second, conscious rats exhibited orienting responses but no signs of anxiety-like behaviours during sustained periods of cl-dSC stimulation in either the open field or EPM.

## Descending pathways mediating autonomic and respiratory responses associated with orienting behaviour

Previous studies from our laboratory demonstrated that autonomic responses to visual, acoustic and somatosensory stimuli, unmasked by disinhibition of the cl-dSC, persist following pre-collicular decerebration (Muller-Ribeiro et al., 2014; Muller-Ribeiro et al., 2016). These observations indicate that the autonomic and respiratory effects of dSC stimulation may be mediated via a previously unrecognized descending pathway to brainstem autonomic and respiratory nuclei. Viral anterograde tracing from cl-dSC neurons defined two main descending medullary bundles: a major ventral pathway that spanned the GiA and adjacent subnuclei in the ventromedial medulla, and a minor dorsal pathway that targeted the dorsomedial spinal trigeminal nucleus.

In regard to the GiA region, an intersectional viral tracing approach revealed direct synaptic contacts with spinally projecting neurons in this region, including both V2A and non-V2A neurons. Optogenetic stimulation of cl-dSC efferent terminals in the GiA region of the ventromedial medulla mimicked the splanchnic and pressor responses evoked by direct optogenetic stimulation of the cl-dSC, indicating that this disynaptic pathway from the cl-dSC to the spinal cord is capable of mediating such autonomic effects. Consistent with this, the onset latency of evoked splanchnic sympathetic responses to optogenetic stimulation of cl-dSC efferent terminals in the GiA was slightly shorter (by ∼13 ms) to responses evoked by direct stimulation of the cl-dSC itself. Extracellular recordings from spinally projecting GiA neurons, including those whose spontaneous activity correlated with SNA, confirmed excitatory inputs from cl-dSC stimulation with a latency (∼10 ms) that aligns well with the difference between the latencies of sympathoexcitatory responses evoked by cl-dSC *vs*. GiA stimulation (∼13 ms). Finally, intersectional viral tracing failed to identify any other potential relay nuclei between the cl-dSC and spinal cord which may be alternative pathways for the short latency sympathetic responses observed. Taken together, these findings suggest that sympathetic responses to cl-dSC stimulation are principally due to activation of GiA neurons and confirm a role for bulbospinal sympathetic premotor GiA neurons in this pathway. Based on the latency of responses and the results of our anterograde tracing experiments, this pathway is likely disynaptic and unlikely to involve supramedullary sites.

Although previous studies have described a sparse innervation of the RVLM by neurons in the SC (Dempsey et al., 2017; Stornetta et al., 2016), anterograde labelling did not reveal anatomical evidence of direct innervation of the RVLM by cl-dSC output neurons in the present study (data not shown), presumably because RVLM-projecting neurons are located in other regions of the SC. This suggests that RVLM neurons may not contribute to sympathetic responses generated by stimulation of the cl-dSC, although the possibility that RVLM neurons are activated via an indirect pathway cannot be ruled out. Nevertheless, although it is well established that RVLM bulbospinal neurons are a major source of excitatory drive to sympathetic nerves and play a critical role in subserving a wide range of cardiovascular reflexes (Dampney, 2016; Guyenet, 2006), other studies have suggested that they are not primarily responsible for mediating sympathetic responses evoked by psychological stress (Carrive & Gorissen, 2008; Furlong et al., 2014).

In contrast to sympathetic responses, respiratory responses to cl-dSC-GiA terminal stimulation were consistently smaller than those evoked by direct cl-dSC stimulation, indicating the involvement of neurons within and beyond the GiA for full expression of respiratory responses. Potential local mediators of respiratory responses include serotonergic raphe obscurus neurons and non-serotonergic GiA neurons, stimulation of which evokes tachypnoea (Depuy et al., 2011; Verner et al., 2004). Supramedullary contributors to cl-dSC-evoked tachypnoea could include the parabrachial/Kölliker-Fuse complex and periaqueductal grey, stimulation of which increases respiratory rate under anaesthesia (Chamberlin & Saper, 1994; Hayward et al., 2003), both of which were identified as sites of dSC terminal fields in the current study.

## Is the GiA region a major hub for integration of somatomotor and autonomic changes associated with orienting behaviour?

Many previous studies have identified the GiA region as a major hub for descending control of reticulospinal neurons that regulate locomotion and orienting behaviour (Bouvier et al., 2015; Dougherty & Kiehn, 2010; Kim et al., 2017; Perreault & Giorgi, 2019). In particular, studies in the mouse and rat have described descending pathways to the GiA region arising from the SC that mediate orienting and other motor functions (Isa et al., 2020; Redgrave et al., 1987; Usseglio et al., 2020). The present results show that the cl-dSC-GiA pathway also has the capacity to regulate autonomic functions.

Our assumption is that optogenetic stimulation of excitatory dSC inputs to presympathetic neurons, which are abundant in and around the GiA (Aicher et al., 1995; Babic & Ciriello, 2004; Cano et al., 2001; Card et al., 2011; Jansen et al., 1995; Strack et al., 1989; Varner et al., 1992), underlies the sympathoexcitatory responses observed in the current study. This is because spinally projecting neurons were strongly activated by dSC stimulation in all cases, including neurons in which spontaneous activity was correlated with SNA and/or blood pressure pulsatility, which are functional signatures of sympathetic premotor neurons in other autonomic cell groups (RVLM: Brown & Guyenet, 1984; caudal raphe: Pilowsky et al., 1995; paraventricular nucleus: Chen & Toney, 2010; A5: Kanbar et al., 2011). However, a number of caveats to this interpretation should be considered: first, antidromic responses to spinal stimulation (or retrograde labelling by CTB) indicate axonal projection to the thoracic spinal cord, but do not reliably indicate whether such neurons innervate sympathetic preganglionic neurons, spinal interneurons or ventral horn motoneurons. Similarly, whereas cardiac

modulation of GiA neuronal activity and its correlation with sympathetic nerve discharge suggest a vasomotor sympathoexcitatory role, such functional profiles may not be unique to sympathetic premotor neurons and exclude cutaneous vasoconstrictor or non-vasomotor sympathetic outputs from consideration. Based on the tail cooling observed in response to dSC stimulation, such outputs are likely to be a component of the system described here. Thus, although the functional and anatomical properties of recorded neurons are consistent with sympathetic premotor neurons, their identification as such should not be considered definitive.

Second, although AAV1-mediated trans-synaptic Cre-tagging only revealed one monosynaptic pathway linking the dSC to the spinal cord (dSC-GiA), this does not preclude the involvement of polysynaptic pathways that involve other pre-sympathetic groups. Of particular note, we and others have demonstrated innervation of RVLM sympathetic premotor neurons by cells within the GiA region (Card et al., 2011; Dempsey et al., 2017), which could potentially be activated by optogenetic stimulation of dSC neurons or their medullary terminals, contributing to the responses observed here. Notwithstanding these caveats, our observations raise the following question: do cl-dSC projections target separate populations of GiA neurons that drive distinct descending motor and autonomic pathways, or do they target multifunctional GiA neurons that provide divergent excitatory drive to sympathetic and somatic motor pools? The latter is possible, because polysynaptic viral tracing indicates that approximately 20% of bulbospinal neurons in the GiA region provide dual innervation to somatomotor and autonomic targets (Kerman, 2008). In fact, the GiA region contains a much higher proportion of such putative dual function neurons than any other region in the brain (Kerman, 2008). Although we found that respiratory responses evoked by optogenetic stimulation of cl-dSC terminal axons in the GiA region were small, there is evidence that V2A neurons in this region regulate breathing via a direct connection to the pre-Bötzinger complex (Crone et al., 2012). Thus, the GiA region is well suited as a major hub for integration of motor, autonomic and respiratory commands from the SC, but further studies are required to determine the extent to which descending inputs from the SC target multifunctional GiA neurons.

Our study focused on descending projections from the cl-dSC that generate orienting behaviours, but it is well established that defence behaviours are also generated from the SC, in particular its rostromedial part (Isa et al., 2020; Redgrave et al., 1987). The descending projections mediating orienting or defence responses have different pathways in the brainstem, but it is noteworthy that both include terminals in the GiA region. It is therefore possible that neurons in the GiA region that generate

autonomic or respiratory responses may receive inputs from both orienting and defence pathways that originate from different parts of the SC.

## Conclusions

In summary, the results of the current study demonstrate that the cl-dSC co-activates somatomotor and autonomic outputs that mimic those observed during orienting behaviours. These responses are largely reproduced by stimulation of cl-dSC axons in the GiA region, a critical relay nucleus for driving motor responses in SC-mediated orienting. The results of the present study suggest that the autonomic and respiratory responses generated by the caudolateral SC in response to external salient stimuli are mediated, at least in part, by direct descending inputs to select brainstem nuclei, and do not depend upon connections with the forebrain regions (such as the medial prefrontal cortex, amygdala and hypothalamus) that are known to play important roles in generating defensive responses to psychologically stressful stimuli (Bondarenko et al., 2015; Carrive, 1993; Dampney, 2015; McDougall et al., 2004; Mohammed et al., 2016; Ulrich-Lai & Herman, 2009).

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

## Additional information

### Data availability statement

Experimental data are available from the corresponding author upon request.

### Competing interests

None declared.

### Author contributions

S.M., A.K.G., R.A.L.D., J.L.C. and P.G.R.B. contributed to the conception and design of the study, and interpretation of the results. E.L., B.D., C.S., E.M. and A.T. were involved in acquisition and analysis of the data. A.M.A. and C.M.H. provided essential equipment and key reagents. S.M. and R.A.L. drafted the manuscript; S.M., A.K.G., R.A.L.D., P.G.R.B., C.M.H. and A.M.A were involved in revising it critically for important intellectual content. Funding by S.M., A.K.G., R.A.L.D., P.G.R.B. and C.M.H. All authors have read and approved the final version of this manuscript and agree to be accountable for all aspects of the work in ensuring that questions related to the accuracy or integrity of any part of the work are appropriately investigated and resolved. All persons designated as authors qualify for authorship, and all those who qualify for authorship are listed.

### Funding

This work was funded by grants from the National Health & Medical Research Council, Australia (APP2001128, APP1127817) and by the Hillcrest Foundation (IPAP2018/0437).

### Acknowledgements

The authors are grateful to Pascal Carrive for providing access to his precious thermal camera with patience and grace.

Open access publishing facilitated by Macquarie University, as part of the Wiley – Macquarie University agreement via the Council of Australian University Librarians.

### Keywords

arousal, cardiovascular, innate behaviours, sensorimotor integration, sympathetic

### Supporting information

Additional supporting information can be found online in the Supporting Information section at the end of the HTML view of the article. Supporting information files available:

**Statistical Summary Document**
**Peer Review History**

