## [Peer Review History · The Journal of Physiology]

Descending pathways from the superior colliculus mediating autonomic and respiratory effects associated with orienting behaviour

Erin Lynch, Bowen R Dempsey, Christine Saleeba, Eloise Monteiro, Anita Turner, Peter GR Burke, Andrew M Allen, Roger Dampney, Cara Margaret Hildreth, Jennifer Cornish, Ann K Goodchild, and Simon McMullan
DOI: 10.1113/JP283789

Corresponding author(s): Simon McMullan (simon.mcmullan@mq.edu.au)

Review Timeline:

Submission Date:	23-Jun-2022
Editorial Decision:	19-Jul-2022
Resubmission Received:	30-Aug-2022
Editorial Decision:	20-Sep-2022
Revision Received:	11-Oct-2022
Accepted:	14-Oct-2022

Senior Editor: Harold Schultz

Reviewing Editor: Vaughan Macefield

Transaction Report:

Dear Dr McMullan,

Re: JP-RP-2022-283492 "Descending pathways from the superior colliculus mediating autonomic and respiratory effects associated with orienting behaviour" by Erin Lynch, Bowen R Dempsey, Christine Saleeba, Eloise Monteiro, Anita Turner, Peter GR Burke, Andrew M Allen, Roger Dampney, Cara Margaret Hildreth, Jennifer Cornish, Ann K Goodchild, and Simon McMullan

Thank you for submitting your manuscript to The Journal of Physiology. It has been assessed by a Reviewing Editor and by 2 Referees and the reports are copied below.

Please let your co-authors know of the following editorial decision as quickly as possible.

As you will see, in its current form, the manuscript is not acceptable for publication in The Journal of Physiology. In comments to me, the Reviewing Editor expressed interest in the potential of this study, but much work still needs to be done (and this may include new experiments) in order to satisfactorily address the concerns raised in the reports.

In view of this interest, I would like to offer you the opportunity to carry out all of the changes requested in full, and to resubmit a new manuscript using the "Submit Special Case Resubmission for JP-RP-2022-283492..." on your homepage.

We cannot, of course, guarantee ultimate acceptance at this stage as the revisions required are substantial. However, we encourage you to consider the requested changes and resubmit your work to us if you are able to complete or address all changes.

A new manuscript would be renumbered and redated, but the original referees would be consulted wherever possible. An additional referee's opinion could be sought, if the Reviewing Editor felt it necessary. A full response to each of the reports should be uploaded with a new version.

I hope that the points raised in the reports will be helpful to you.

Yours sincerely,

Harold D Schultz
Senior Editor
The Journal of Physiology
<https://jp.msubmit.net>
<http://jp.physoc.org>
The Physiological Society
Hodgkin Huxley House
30 Farringdon Lane
London, EC1R 3AW
UK
<http://www.physoc.org>
<http://journals.physoc.org>

EDITOR COMMENTS

Reviewing Editor:

Dear Dr McMullan

Thank you for submitting your manuscript to The Journal of Physiology. I have now received the reports from two experts in the field, both of whom see merit in your study but - as you will see from their detailed reports - raise concerns about how your data are presented and interpreted. I invite you to take these points onboard and to submit a new version of your manuscript. I will endeavour to use these same reviewers when you resubmit your new manuscript, so please address their concerns.

Senior Editor:

Comments for Authors to ensure the paper complies with the Statistics Policy:

Carefully adhere to the journals data and statistical reporting guidelines. Must state actual p values: SEM not allowed.

Carefully adhere to journal guidelines for describing use of animals and anesthesia procedures.

REFeree COMMENTS

Referee #1:

The authors offer anatomical and physiological evidence supporting the possibility that the deep layers of the superior colliculus (SC) contain neurons that activate the sympathetic nervous system via bulbospinal neurons located in the gigantocellular nucleus pars alpha (GiA). This pathway is assumed to contribute to the autonomic correlates of the orienting responses generated by activation of the SC.

The work is original, the experiments are well executed and the report is nicely illustrated. The data are fine and the interpretations generally sensible. This is overall an excellent research report.

However, one could take issue with the electrophysiological identification of the GiA "cardiovascular neurons" and some of the conclusions derived from these experiments. Here are the reasons.

The reference to AD Loewy's work with the pseudorabies virus (bottom of page 20) seems to imply that, the GiA contains "presympathetic" neurons i.e. neurons directly antecedent to sympathetic preganglionic neurons. This is certainly one interpretation but not the only one. The retrograde propagation of PRV does not stop at the first neuron, Loewy's GiA neurons could be upstream from presympathetic neurons. Also, antidromic activation using an electrode located in the dorsolateral funiculus does not provide compelling evidence that the backfired neurons innervate the IML, especially without performing very careful depth-threshold curves to show that the lowest threshold of AD activation is located in the IML (cf. work by Barman and Gebber in the 80s). Pulse modulation of unit discharge is likewise not a compelling piece of evidence that the neurons are regulating the sympathetic system. P Dell in the 1950s had already demonstrated that baroreceptor stimulation could cause skeletal muscle atonia. In addition, the authors have not demonstrated that the AD-activated neurons respond negatively to stimulation of arterial baroreceptors. In short, although the experiments are adequate, the authors should consider alternate interpretations. For example, the possibility that the bulbospinal GiA neurons target the SPGNs via spinal interneurons is not ruled out by the CTB experiments; CTB cannot be confined to the IML and presympathetic interneurons can be in close proximity to the IML. Also, the authors show that the SC directly innervate the A5 neurons (Figure 5); Most A5 neurons innervate the IML and regulate predominantly the splanchnic nerve, a focus of the present study. Finally, GiA contains serotonin and/ or VGlut3 bulbospinal neurons that regulate tail blood flow, many of which are demonstrably presympathetic. By the way are these neurons CHX10-immunoreactive? These serotonin and/ or VGlut3 bulbospinal neurons could also mediate some of the autonomic effects of the SC.

In short, it is recommended that the authors define what they mean by "cardiovascular neurons" and modify the discussion to offer a broader range of possible interpretations as to how the autonomic nervous system is activated by the SC.

Minor Details:

Key resource Table: atipemazole and bupivacaine are misspelt. Viral vectors: please define the unit (viral genome per cc?)

Page 13: "300 microns below the facial motor nucleus should be the ventral surface of the brainstem. Is this what you mean?"

Referee #2:

Summary

The authors use optogenetic and viral tracing approaches to demonstrate that the GiA may underlie, at least in part, the autonomic and respiratory responses evoked by light-activation of the dSC. While the study design is straightforward and the techniques are difficult and costly to implement, the manuscript suffers from inadequate reporting of the methods and results and inconsistencies between the results and their interpretation.

Major Concerns

P2, Key Points #2: While the authors do describe the orienting responses observed, no data is shown throughout the

manuscript. Instead, the authors point to previous work and report the similarity in their observations, which is fine considering that the central point of the manuscript concerns the descending pathways that mediate autonomic and respiratory responses during orienting behaviors. However, inclusion of these data in the manuscript to improve the clarity of the results with respect to the previous literature. For instance, while the authors observed only contraversive head-only orienting turns, Isa et al. (2020) reported observing either ipsiversive, contraversive or upward head movements during head-only orienting turns and only contraversive body-turns when activating the crossed dSC pathway. Similarly, Usseglio et al. (2020) observed only ipsilateral orienting turns when optogenetically activating the GiA.

Was contraversive turning present for both left and right cl-DSC injection sites? Were left and right cl-DSC injections even performed? In the data analysis section, there is some reference to the analysis of left dSC injections, but the specifics of where the injections were made were not described in the experimental methods sections.

Fig 1B2: Which portion of nose velocity is significant? Why was only the significance of the interaction term reported, rather than comparing the velocity during stimulation versus baseline, or a specific time-point versus baseline? The interaction term would imply only that at some point in the recording the measurement was different---this could be at baseline, during the stimulation or after the stimulation.

Was this velocity increase appearing during circling behaviors or during straight, fleeing-like behaviors? Perhaps a plot of the trajectories, averaged or representative, could aid in the interpretation of these data.

Fig. 1B1: In the methods section, there is some reference to counting the number of rightward circles after left dSC injections. I assume this is what is plotted here. Was there circling in the other direction? Were injections in the right dSC ever assessed?

Fig. 3A1: Was the increase breathing frequency sniffing? Perhaps a comparison of spontaneous sniffing would aid the interpretation of this respiratory response to optogenetic activation of the dSC.

Fig. 3A3: EEG Frequency associated with significant difference in power? What is the relevance for measuring the theta rhythm with respect to changes in cardio-respiratory variables? Does the activation of a theta rhythm in the forebrain drive the brainstem cardio-respiratory network? The rationale for this measurement should be explained at some point in the manuscript. Or, if it is purely tangential to the central question of the manuscript (what is the circuit that underlies cardio-respiratory changes during orienting behaviors), it should be removed.

Again, the interaction term is not appropriate to statistically describe the optogenetic-evoked effects. From the power spectrum, it is clear that this is largely driven by a peak at ~5Hz, but there are also differences at other frequencies. Perhaps a statistical comparison of delta, theta, and gamma-band power would be beneficial to clearly communicate the results.

Fig. 3B3: Time-points at which tail temperature is significantly different? Again, the interaction term could be sensitive the rebound vasodilation of the tail that is evident in the raw traces. The data should be compared between baseline, during stimulation and after stimulation.

Fig. 4E & F: Time-points at which changes in SNA and sAP are significantly different from control?

Fig. 3& 4: Latency for recovery of respiratory and autonomic variables to baseline? What causes lasting changes in respiratory and autonomic arousal?

Fig. 4: Previous results from this group (e.g., Muller-Rebeiro et al., 2014) showed profound effects on SNA and the respiratory pattern that were evoked by naturalistic stimuli in anesthetized animals after local disinhibition of the dSC. The results observed in this figure in an identical preparation, but evoked by optogenetic activation of the dSC, are far less convincing. In the previous work, claps or light stimuli evoked large synchronized bursts of SNA that also suppressed tonic SNA, whereas in the present study, only a mild increase in tonic SNA was evoked. This raises the question of whether optogenetic activation of the dSC is even capable of evoking an orienting response under anesthesia without additional auditory or visual inputs. Did the authors ever pair naturalistic input with ChR2-activation of the dSC? This comparison would have been very helpful to interpret the results. In any case, a discussion of the discrepancies between this and the authors' previous work is necessary.

Fig. 4: AP in A & B: Why is the diastolic pressure 50 mmHg? This seems un-physiologically low. How was the pressure transducer calibrated?

Fig. 4: SNA in A & B: The long time-scale of the traces makes it difficult to appreciate whether this is a good recording of SNA or just noise, especially in the representative control experiment.

Fig. 4C Changes in T_i , T_e ? Is there a change in respiratory pattern? The traces shown do not appear to be sniffing, but this is difficult to assess since changes in respiratory frequency are presented as percent of control. What are the absolute changes in frequency? Is increased inspiratory duration significant? Or is the increase in respiratory frequency due to changes in the expiratory components of the network? The latter would go against the later speculation in the Discussion that respiratory frequency changes are mediated by the pre-Botzinger complex (p. 26).

Fig. 5E: While the high-power images clearly show spinally-projecting neurons that are densely innervated by cl-dSC, this pattern does not appear uniform in the GiA in the low power images. The interpretation of these critical data could be improved by also showing some representative images where there is less dense innervation of the GiA by cl-dSC inputs.

Fig. 6A: It was not reported in the Methods how the location of the fiber optic cannula was assessed, nor can it be inferred from the histologic image of ChR2-YFP expression.

Fig. 7: Significant difference between GiA terminal field- and cl-dSC-ChR2 stimulation was only observed for respiratory frequency. Why is this not shown in representative experiment, e.g. Fig. 7A?

Fig. 8D & E: The unit in D does not appear to be a putative sympathetic premotor unit (as in E). Were putative sympathetic premotor units equally strongly activated by dSC light stimulation? It was already known that there are direct connections between dSC and GiA (Usseglio et al., 2020)

p. 24: "Viral anterograde tracing from cl-dSC neurons defined two main descending pathways - a major ventral pathway to the GiA region in the ventromedial medulla, and a minor dorsal pathway that targeted the dorsomedial spinal trigeminal nucleus." In the Results section, the authors identify putative projections of the dSC to the locus coeruleus, parabrachial/Kolliker-Fuse regions, GiA dorsal, lateral paraventricular reticular nucleus, GiA ventral, raphe obscurus, raphe pallidus and raphe magnus. In the legend to Fig 5, it is additionally noted that the dorsomedial spinal trigeminal nucleus is innervated by the dSC. Thus, it is very confusing that in the Discussion, the authors condense the dSC projections into only two pathways, especially considering that the key physiologic experiment to justify their assertion that the GiA is a key relay to mediate autonomic and respiratory effects of dSC activation---the activation of dSC ChR2+ fibers in the GiA---did not fully replicate the effect of dSC light activation.

p. 25: "intersectional viral tracing failed to identify any other potential relay nuclei between the cl-dSC and spinal cord which may be alternative pathways for the short latency sympathetic responses observed."

Only one section from one animal is shown in Fig 5F to justify this statement. However, even in that single section, there is

clearly GFP+ cells beyond the narrow borders of the GiA. These data should be more thoroughly reported to justify the authors' conclusion.

p. 25: "Although previous studies have described a sparse innervation of the rostral ventrolateral medulla (RVLM) by neurons in the SC (Stornetta et al., 2016; Dempsey et al., 2017), anterograde labelling did not reveal anatomical evidence of direct innervation of the RVLM by cl-dSC output neurons in the present study,..." The only data presented to justify this statement is Fig. 5Biv, which appears near the level of the RVLM (it is difficult to be certain without a Nissl counterstain), with positive signal in its ventrolateral extents. Moreover, the legend states "Labelled fibres were infrequently encountered in the caudal medulla (Biv)." These contradictory data and descriptions of the results are incompatible with their interpretation in the Discussion. Clarification with a more thorough presentation of this difficult experiment would benefit the manuscript.

p. 27: "The results of the present study suggest that the behavioural, autonomic and respiratory responses generated by the caudolateral superior colliculus in response to external salient stimuli are mediated, at least in part, by direct descending inputs to select brainstem nuclei, and do not depend upon connections with forebrain regions" The conclusion that the "behavioural, autonomic and respiratory responses" depend on "direct descending inputs to select brainstem nuclei" is confusing given that the study is entirely focused on whether the GiA mediates autonomic and respiratory responses.

Minor Concerns

p.15-16: The description of the various data analyses made are out of order with respect to the presentation of the data in the Figures and Results sections. It would be helpful to organize these sections in the same order to aid in understanding what was actually done.

Fig. 6, legend: "example illustrating cannula position" should be "[schematic] illustrating cannula position"

Fig. 8D, legend: Bottom panel: What is blue? What is gray? Perhaps separating the PSTH from the raster plot would yield a more legible figure?

p.16: What was the concentration of hexamethonium bromide used to determine the noise level in SNA recordings? Its dosage is currently listed as 5mg, which is unclear.

ADDITIONAL FORMATTING REQUIREMENTS FOR RESUBMISSION:

-Author photo and profile. First (or joint first) authors are asked to provide a short biography (no more than 100 words for one author or 150 words in total for joint first authors) and a portrait photograph. These should be uploaded and clearly labelled with the revised version of the manuscript. See Information for Authors for further details.

-Your manuscript must include a complete Additional Information section

-Please upload separate high-quality figure files via the submission form.

-Please ensure that the Article File you upload is a Word file.

-A Statistical Summary Document, summarising the statistics presented in the manuscript, is required upon revision. It must be on the Journal's template, which can be downloaded from the link in the Statistical Summary Document section here: https://jp.msubmit.net/cgi-bin/main.plex?form_type=display_requirements#statistics

-Papers must comply with the Statistics Policy https://jp.msubmit.net/cgi-bin/main.plex?form_type=display_requirements#statistics

In summary:

-If $n \leq 30$, all data points must be plotted in the figure in a way that reveals their range and distribution. A bar graph with data points overlaid, a box and whisker plot or a violin plot (preferably with data points included) are acceptable formats.

-If $n > 30$, then the entire raw dataset must be made available either as supporting information, or hosted on a not-for-profit repository e.g. FigShare, with access details provided in the manuscript.

- n clearly defined (e.g. x cells from y slices in z animals) in the Methods. Authors should be mindful of pseudoreplication.

-All relevant n values must be clearly stated in the main text, figures and tables, and the Statistical Summary Document (required upon revision)

-The most appropriate summary statistic (e.g. mean or median and standard deviation) must be used. Standard Error of the Mean (SEM) alone is not permitted.

-Exact p values must be stated. Authors must not use 'greater than' or 'less than'. Exact p values must be stated to three significant figures even when 'no statistical significance' is claimed.

-Statistics Summary Document completed appropriately upon revision

-A Data Availability Statement is required for all papers reporting original data. This must be in the Additional Information section of the manuscript itself. It must have the paragraph heading "Data Availability Statement". All data supporting the results in the paper must be either: in the paper itself; uploaded as Supporting Information for Online Publication; or archived in an appropriate public repository. The statement needs to describe the availability or the absence of shared data. Authors must include in their Statement: a link to the repository they have used, or a statement that it is available as Supporting Information; reference the data in the appropriate section(s) of their manuscript; and cite the data they have shared in the References section. Whenever possible the scripts and other artefacts used to generate the analyses presented in the paper should also be publicly archived. If sharing data compromises ethical standards or legal requirements then authors are not expected to share it, but must note this in their Statement. For more information, see our Statistics Policy.

-Please include an Abstract Figure. The Abstract Figure is a piece of artwork designed to give readers an immediate understanding of the research and should summarise the main conclusions. If possible, the image should be easily 'readable' from left to right or top to bottom. It should show the physiological relevance of the manuscript so readers can assess the importance and content of its findings. Abstract Figures should not merely recapitulate other figures in the manuscript. Please try to keep the diagram as simple as possible and without superfluous information that may distract from the main conclusion(s). Abstract Figures must be provided by authors no later than the revised manuscript stage and should be uploaded as a separate file during online submission labelled as File Type 'Abstract Figure'. Please ensure that you include the figure legend in the main article file. All Abstract Figures should be created using BioRender. Authors should use The Journal's premium BioRender account to export high-resolution images. Details on how to use and access the premium account are included as part of this email.

Author Response

Referee #1:

The authors offer anatomical and physiological evidence supporting the possibility that the deep layers of the superior colliculus (SC) contain neurons that activate the sympathetic nervous system via bulbospinal neurons located in the gigantocellular nucleus pars alpha (GiA). This pathway is assumed to contribute to the autonomic correlates of the orienting responses generated by activation of the SC.

The work is original, the experiments are well executed and the report is nicely illustrated. The data are fine and the interpretations generally sensible. This is overall an excellent research report. However, one could take issue with the electrophysiological identification of the GiA "cardiovascular neurons" and some of the conclusions derived from these experiments. Here are the reasons.

The reference to AD Loewy's work with the pseudorabies virus (bottom of page 20) seems to imply that, the GiA contains "presympathetic" neurons i.e. neurons directly antecedent to sympathetic preganglionic neurons. This is certainly one interpretation but not the only one. The retrograde propagation of PRV does not stop at the first neuron, Loewy's GiA neurons could be upstream from presympathetic neurons. Also, antidromic activation using an electrode located in the dorsolateral funiculus does not provide compelling evidence that the backfired neurons innervate the IML, especially without performing very careful depth threshold curves to show that the lowest threshold of AD activation is located in the IML (cf. work by Barman and Gebber in the 80s). Pulse modulation of unit discharge is likewise not a compelling piece of evidence that the neurons are regulating the sympathetic system. P Dell in the 1950s had already demonstrated that baroreceptor stimulation could cause skeletal muscle atonia. In addition, the authors have not demonstrated that the AD-activated neurons respond negatively to stimulation of arterial baroreceptors.

In short, although the experiments are adequate, the authors should consider alternate interpretations. For example, the possibility that the bulbospinal GiA neurons target the SPGNs via spinal interneurons is not ruled out by the CTB experiments; CTB cannot be confined to the IML and presympathetic interneurons can be in close proximity to the IML. Also, the authors show that the SC directly innervates the A5 neurons (Figure 5); Most A5 neurons innervate the IML and regulate predominantly the splanchnic nerve, a focus of the present study. Finally, GiA contains serotonin and/ or VGlut3 bulbospinal neurons that regulate tail blood flow, many of which are demonstrably presympathetic. By the way are these neurons CHX10-immunoreactive? These serotonin and/ or VGlut3 bulbospinal neurons could also mediate some of the autonomic effects of the SC.

In short, it is recommended that the authors define what they mean by "cardiovascular neurons" and modify the discussion to offer a broader range of possible interpretations as to how the autonomic nervous system is activated by the SC.

Author Response: *We thank the Reviewer for their kind words and thoughtful and constructive comments. They make an excellent point regarding the identification of putative presympathetic neurons recorded in the GiA; we agree that careful consideration of the technical limitations of the experiment and alternative interpretations is warranted.*

Action Taken: *We have edited the manuscript to address the issues raised by Reviewer 1. We have included relevant references that demonstrate likely monosynaptic projections between GiA neurons and sympathetic preganglionic neurons (p 21: Aicher et al., 1995, Babic & Ciriello, 2004) and have extensively expanded the Discussion (p. 27) to acknowledge the technical limitations of the antidromic stimulation/ retrograde tracing/ functional identification, and have included consideration of an alternative pathway through the RVLM, which we think is most likely based on the evidence.*

Minor Details:

1. Key resource Table: atipemazole and bupivacaine are misspelt.

Author Response: *Amended*

2. Viral vectors: please define the unit (viral genome per cc?)

Author Response: *Amended*

3. Page 13: "300 microns below the facial motor nucleus should be the ventral surface of the brainstem. Is this what you mean?"

Author Response: *Clarified in the text – 300 μm deep to the position of the largest field potential, is just dorsal to the ventral surface (at the lateral medulla) and corresponds to a position a few hundred microns dorsal to the pyramid.*

Referee #2:

Summary

The authors use optogenetic and viral tracing approaches to demonstrate that the GiA may underlie, at least in part, the autonomic and respiratory responses evoked by light-activation of the dSC. While the study design is straightforward and the techniques are difficult and costly to implement, the manuscript suffers from inadequate reporting of the methods and results and inconsistencies between the results and their interpretation.

Major Concerns

1. P2, Key Points #2: While the authors do describe the orienting responses observed, no data is shown throughout the manuscript. Instead, the authors point to previous work and report the similarity in their observations, which is fine considering that the central point of the manuscript concerns the descending pathways that mediate autonomic and respiratory responses during orienting behaviors. However, inclusion of these data in the manuscript to improve the clarity of the results with respect to the previous literature. For instance, while the authors observed only contraversive head-only orienting turns, Isa et al. (2020) reported observing either ipsiversive, contraversive or upward head movements during head-only orienting turns and only contraversive body-turns when activating the crossed dSC pathway.

Similarly, Usseglio et al. (2020) observed only ipsilateral orienting turns when optogenetically activating the GiA. Was contraversive turning present for both left and right cl-DSC injection sites? Were left and right cl-DSC injections even performed? In the data analysis section, there is some reference to the analysis of left dSC injections, but the specifics of where the injections were made were not described in the experimental methods sections.

Author Response: *We thank the Reviewer for pointing out this omission and have attempted to clarify these details in the revised manuscript, which now includes additional data and a new figure. In the experiments described we only ever saw contraversive movements, which were qualitatively very similar to those shown by Isa et al (2020). When piloting experiments we included animals instrumented on both left and right hemispheres (and observed only contraversive responses irrespective of side stimulated). However, our electrophysiology setup is optimised for recordings from the left splanchnic nerve, so for our final cohort we limited injections to the left dSC so that nerve recordings were made on the same side as the predominantly ipsilateral GiA terminal field. Recordings of GiA neurons were restricted to the left side for the same reason.*

Action Taken: *We have updated the Methods section to clarify that AAV injections were restricted to the left dSC and have explicitly stated that no ipsilateral movement was ever noted. We have updated Figure 1 to include a new panel that shows an annotated video frame demonstrating behavioural responses in a representative experiment.*

2. Fig 1B2: Which portion of nose velocity is significant? Why was only the significance of the interaction term reported, rather than comparing the velocity during stimulation versus baseline, or a specific time-point versus baseline? The interaction term would imply only that at some point in the recording the measurement was different---this could be at baseline, during the stimulation or after the stimulation.

Author Response: *The Reviewer writes “The interaction term would imply only that at some point in the recording the measurement was different---this could be at baseline, during the stimulation or after the stimulation”. We don’t agree with this interpretation; rather, in the context of the current experimental paradigm, a significant 2-way ANOVA interaction tells us that, viewed holistically, the two experimental groups (control vs. ChR2) respond differently to a stimulus (dSC stimulation) over time. This remains true whether or not significant differences are identified when the data are compared against each other point by point using post-tests.*

We have selected this approach throughout for the analysis of parameters with complex dynamics because we believe it represents the raw experimental data more faithfully and makes the analysis less arbitrary than other approaches. For example, in Figure 1B2, which plots nose movement before, during and after dSC stimulation, the plotted data show a clear but short-lived increase in movement in the first few minutes of stimulation which accorded with observations in the laboratory and, for reasons we do not understand, seems to largely recede for the remainder of the stimulation period. This representation of the data captures the complex dynamics of the experiment more faithfully than would be the case of we merged activity into a simple averages of baseline vs. stimulation.

It is however true that post-tests are required to unambiguously identify time-points at which pairings are significantly different. However, given the relatively low statistical power of post-tests compared to the 2-way ANOVA, we recognised that inclusion of such data were likely to be of limited use, potentially distracting readers and complicating figures. More importantly, we concluded that such data would provide no useful guidance for the interpretation of the data.

To illustrate this point, the figure below graphs the effects of dSC stimulation on tail temperature (same data as the heat map shown in Figure 3B) showing a relative drop in tail temperature during stimulation and a rebound warming in the minutes after stimulation in ChR2 vs. Control rats. To this we have also added indicators of statistical significance (the results of post-tests at each time point) in green beneath the main graph. We feel this additional data does little to aid interpretation of the data and are, at worst, distracting.

We feel strongly that such data do not provide any additional biological insights– the take-home message here and in other similar graphs throughout our paper is that dSC stimulation evokes measurable effects on blood pressure, sympathetic nerve activity, respiratory rate, tail temperature and animal movement in dynamic and complex ways.

3. Was this velocity increase appearing during circling behaviors or during straight, fleeing-like behaviors? Perhaps a plot of the trajectories, averaged or representative, could aid in the interpretation of these data.

Author Response: *The transient increase in velocity is interesting; it occurred during circling behaviours (we never saw straight flight-like behaviours in response to dSC stimulation). Rather than representing increased velocity of movement (i.e. running rather than walking), we think the change in this parameter likely reflects an increased probability of movement vs. immobility (i.e. stimulation makes rats more likely to move around than sit still). It's maybe better conceptualised as increased distance travelled within the period of observation rather than an increase in velocity per se.*

We've thought carefully about how to best represent these data; in a previous draft we included a figure that mapped tracking from representative experiments (below). These data show that 1 Hz stimulation does not elicit any obvious change in gross locomotor behaviour – rats do not freeze, run in circles, or restrict their movement to proximity to the cage wall, and still engage in exploratory behaviour (periods in which rats stand on their back legs and poke their noses over the cage sides are indicated by dots that fall along or beyond the box boundary). Instead, careful examination of the middle panel of the ChR2 rat reveals swirling patterns that correspond to circling during stimulation and an increase in the total length of the blue path, which corresponds to the total distance travelled. However, we felt that these graphs are still hard to interpret; we concluded that they may prove distracting and opted to omit this figure from the current submission. We'd value the Reviewer's perspective on whether a revised version would benefit from inclusion of this graphic.

Action Taken: We have relabelled Figure 1C2 as 'Distance Travelled' to reinforce the take-home message.

4. Fig. 1B1: In the methods section, there is some reference to counting the number of rightward circles after left dSC injections. I assume this is what is plotted here. Was there circling in the other direction? Were injections in the right dSC ever assessed?

Author Response: See response to #1.

5. Fig. 3A1: Was the increase breathing frequency sniffing? Perhaps a comparison of spontaneous sniffing would aid the interpretation of this respiratory response to optogenetic activation of the dSC.

Author Response: Very interesting question. Sniffing is of course an integrated behaviour that consists of vibrissal whisking, retraction and protraction of the snout, and head movements in addition to bursts of very rapid inspiratory activity, which occur at 5- 11 Hz in the rat (reviewed by Deschênes et al. (2012). Sniffing and whisking in rodents. *Curr Opin Neurobiol*, 22(2), 243-250. doi:10.1016/j.conb.2011.11.013). Nalivaiko et al. (2012, *Front Physiol*, 2(114) doi:10.3389/fphys.2011.00114) recorded bursts of very rapid breaths (6-10 Hz) in response to salient naturalistic acoustic stimuli, which they classified as sniffs. However, several observations suggest the increases in breathing rate evoked by optogenetic stimulation of dSC, reported here, resulted from stimulation of the central respiratory central pattern generator (CPG) rather than sniffing. Firstly, under anaesthesia, direct stimulation of the dSC evoked an increase in breathing frequency in the absence of sniffing that was stimulus-dependent (Figure 4; 10Hz vs 20 Hz optical stimulation). Second, optical stimulation of dSC in free behaviour also increased breathing frequency in a stimulus-dependent manner (Figure 3), suggesting an underlying central mechanism conserved across anaesthetised and conscious states. We cannot rule out behaviour indirectly influencing respiratory increases, such as sniffing.

Certainly, dSC-evoked increases in respiratory frequency were much more variable in free-behaviour, and the maximum respiratory frequency observed during optogenetic dSC stimulation (~6 Hz) overlaps with the low end of the sniffing frequency range (5-11 Hz). However, such high respiratory

rates were not sustained for the duration of responses (the average respiratory frequency during stimulation was ~3.5 Hz). We therefore don't think that responses were rapid enough, or reliably reproduced with each optical stimulation, to support behavioural sniffing.

Action taken: *The Reviewer has highlighted the importance of including the raw respiratory parameters to the text; these have now been added to the Results (p.19).*

6. Fig. 3A3: EEG Frequency associated with significant difference in power? What is the relevance for measuring the theta rhythm with respect to changes in cardio-respiratory variables? Does the activation of a theta rhythm in the forebrain drive the brainstem cardiorespiratory network? The rationale for this measurement should be explained at some point in the manuscript. Or, if it is purely tangential to the central question of the manuscript (what is the circuit that underlies cardio-respiratory changes during orienting behaviors), it should be removed.

Again, the interaction term is not appropriate to statistically describe the optogenetic-evoked effects. From the power spectrum, it is clear that this is largely driven by a peak at ~5Hz, but there are also differences at other frequencies. Perhaps a statistical comparison of delta, theta, and gamma-band power would be beneficial to clearly communicate the results.

Author Response: *Shifts in the EEG spectrum, with a concentration of power around theta rhythm, are a hallmark of orienting responses to ecologically salient stimuli, commonly interpreted as an index of arousal, engagement with the environment, or intention to move when recorded in awake animals. In this context, and in light of the absence of measurable anxiogenic effects of dSC stimulation (elevated plus, open field tests, ultrasonic vocalisation), EEG recordings provide an objective indicator that the behavioural responses observed are a component of an integrated orienting response, rather than a simple motor effect.*

However, we accept that, in this case, statistical comparison of the individual frequency components of the spectrogram may assist interpretation of the data.

Action Taken: *We have broken the EEG into delta, theta, alpha & beta components (gamma frequencies are too high to measure in this setup) and have updated Figure 3 to include a new panel that depicts the EEG power for the four main bands with statistical comparisons as described – as expected, we report a decline in delta power, an increase in theta, and no effect alpha or beta power. We have also modified the results and methods text to include the EEG power analysis by band. Thank you for this suggestion.*

7. Fig. 3B3: Time-points at which tail temperature is significantly different? Again, the interaction term could be sensitive the rebound vasodilation of the tail that is evident in the raw traces. The data should be compared between baseline, during stimulation and after stimulation.

Author Response: *See response to #2 above; we hope you agree that in this specific case the post-test data don't really help.*

8. Fig. 4E & F: Time-points at which changes in SNA and sAP are significantly different from control?

Author Response: See response to #2 above; identifying particular data points that are independently different from one another does not provide any biologically useful insights; the key point is that SNA/sAP respond differently to SC illumination in Chr2 and Control rats.

9. Fig. 3& 4: Latency for recovery of respiratory and autonomic variables to baseline? What causes lasting changes in respiratory and autonomic arousal?

Author Response: It's true that, in awake rats, dSC-evoked increases in respiratory rate took a few seconds to recover after cessation of stimulation (Figure 3A, top panel). We note that this latency to recovery disappears under anaesthesia (Figure 4A, top panel). We don't have any definitive explanation for this observation, but speculate that in conscious animals it is likely that multiple mechanisms overlap: direct stimulation of the respiratory CPG, superimposed by respiratory activity associated with orientation/exploration/vigilance, could all coalesce to drive changes that take a few seconds to fully disengage.

We don't think there is any evidence of long-lasting autonomic effects; the effect of dSC stimulation on tail temperature, characterised by tail-cooling during stimulation followed by tail-warming in the post-stimulation period (Figure 3B) is easily explained: tail cooling results in heat retention (probably aided by brown adipose thermogenesis, although this was not measured) and a likely increase in core temperature over the stimulus period. This excess heat is dissipated in the post-stimulation period as a result of homeostatic vasodilation, shunting blood to the tail, resulting in the observed increase in temperature. Similar effects have been reported by Carrive and colleagues in response to psychological stress (Vianna et al. (2008). *Neuroscience*, 153(4), 1344-1353. doi:10.1016/j.neuroscience.2008.03.033). Certainly, under anaesthesia, sympathetic and cardiovascular responses to dSC stimulation immediately recover after stimulation (Figure 4E & F).

10. Fig. 4: Previous results from this group (e.g., Muller-Rebeiro et al., 2014) showed profound effects on SNA and the respiratory pattern that were evoked by naturalistic stimuli in anesthetized animals after local disinhibition of the dSC. The results observed in this figure in an identical preparation, but evoked by optogenetic activation of the dSC, are far less convincing. In the previous work, claps or light stimuli evoked large synchronized bursts of SNA that also suppressed tonic SNA, whereas in the present study, only a mild increase in tonic SNA was evoked. This raises the question of whether optogenetic activation of the dSC is even capable of evoking an orienting response under anesthesia without additional auditory or visual inputs. Did the authors ever pair naturalistic input with Chr2-activation of the dSC? This comparison would have been very helpful to interpret the results. In any case, a discussion of the discrepancies between this and the authors' previous work is necessary.

11. **Author Response:** In our previous study (Müller-Ribeiro et al., 2014) the dSC was activated by naturalistic stimuli under conditions where tonic GABAergic inhibition of the dSC was blocked, whereas in the present study dSC neurons were activated by optogenetic stimulation in the presence of tonic GABAergic inhibition. Therefore, the experimental conditions in the two

cases were quite different, and we do not think there is necessarily any discrepancy between the two sets of observations. Furthermore, we do not see how pairing naturalistic stimuli with optogenetic gain-of-function will help the interpretation of the results. RFig. 4: AP in A & B: Why is the diastolic pressure 50 mmHg? This seems un-physiologically low. How was the pressure transducer calibrated?

Author Response: We have checked the raw recordings for both traces – we don't think there is any cause for concern; Please note that the systolic pressure is >120 mmHg in both cases, with a large pulse pressure of 50-60 mmHg, within the normal physiological range. The pressure amplifier has an integrated calibration feature that passes a known voltage through the amplifier circuit, which is used to mitigate against drift before every experiment. We use clinical neonatal pressure transducers; the calibration pulse and pressure transducer are calibrated to a sphygmomanometer every few weeks. In over ten years using this setup (changing the pressure transducers pretty regularly) we have never encountered significant drift – typically <5 mmHg.

12. Fig. 4: SNA in A & B: The long time-scale of the traces makes it difficult to appreciate whether this is a good recording of SNA or just noise, especially in the representative control experiment.

Author Response: The timescales are optimised so that readers can clearly identify the stable baseline parameters, the responses to optical stimulation, and recovery to baseline. We feel that our readers can judge the quality of SNA recording in Figure 4 as currently presented, especially the examples of the optically evoked SNA in 4A, D, E. This is an already dense figure, and we'd prefer not to add extra panels to this figure to highlight characteristics of baseline sympathetic vasomotor activity.

13. Fig. 4C Changes in T_i , T_e ? Is there a change in respiratory pattern? The traces shown do not appear to be sniffing, but this is difficult to assess since changes in respiratory frequency are presented as percent of control. What are the absolute changes in frequency? Is increased inspiratory duration significant? Or is the increase in respiratory frequency due to changes in the expiratory components of the network? The later would go against the later speculation in the Discussion that respiratory frequency changes are mediated by the pre-Bötzinger complex (p. 26).

Author Response: Diaphragm EMG recordings are not typically used for examining respiratory patterning parameters (T_i , T_e , T_i/T_{tot} etc). We commonly perform these types of analyses with neurogram recordings of phrenic and vagal activity (References: Toor S 2019; Burke PG 2010). These recordings and analysis were not required for the current study.

14. Fig. 5E: While the high-power images clearly show spinally-projecting neurons that are densely innervated by cl-dSC, this pattern does not appear uniform in the GiA in the low power images. The interpretation of these critical data could be improved by also showing some representative images where there is less dense innervation of the GiA by cl-dSC inputs.

Author Response: Thank you for this interesting suggestion. The paper already features three low power images that capture this terminal field: in addition to the example cited (Figure 5E), Figure 5B charts the entire medullary distribution of labelling in a different experiment, while Figure 6A, shows cl-dSC->GiA terminal labelling with Chr2-YFP. We feel that a sufficient diversity of low-power images is presented in the manuscript, accurately representing the both the density and distribution of the field.

15. Fig. 6A: It was not reported in the Methods how the location of the fiber optic cannula was assessed, nor can it be inferred from the histologic image of Chr2-YFP expression.

Author Response: Thank you – that’s a good pickup. This has been amended – detail added to Methods, final paragraph of section “Physiological effects of dSC stimulation”

Author Response: Resolved.

16. Fig. 7: Significant difference between GiA terminal field- and cl-dSC-ChR2 stimulation was only observed for respiratory frequency. Why is this is not shown in representative experiment, e.g. Fig. 7A?

Author Response: Figure 7 shows pooled data comparing blood pressure, sympathetic, respiratory and heart rate responses to dSC and GiA stimulation, but no example traces from a representative experiment. The pooled data show what the Reviewer has suggested; no significant differences between sympathetic or systolic blood pressure responses (2-way ANOVA $P > 0.99!$), no significant difference in effects on HR (no effect either way) or inspiratory burst amplitude, but a diminished tachypnoeic effect of GiA stimulation compared to dSC stimulation on respiratory cycle frequency. These are shown in the representative traces: dSC stimulation (Figure 4A) increased respiratory frequency from ~1 Hz to around 1.5 Hz, whereas GiA stimulation (Figure 6B) increases respiratory frequency from ~1.2 to ~1.4 (i.e. a diminished effect on respiratory frequency of GiA stimulation).

17. Fig. 8D & E: The unit in D does not appear to be a putative sympathetic premotor unit (as in E). Were putative sympathetic premotor units equally strongly activated by dSC light stimulation? It was already known that there are direct connections between dSC and GiA (Usseglio et al., 2020)

Author Response: Figure 8, Panel D shows neural (lower panel, raster/PSTH) and simultaneously recorded sympathetic (green trace, upper panel) responses to low frequency optogenetic dSC stimulation, but provides no indicator of whether the neuron is presympathetic or not – one can only conclude that both the neuron and the sympathetic nerve respond to 0.5 Hz dSC stimulation, and that the neural responses occurred before the sympathetic response. We did not notice any difference in responsiveness of likely pre-sympathetic neurons compared to neurons in which pulse modulation or spike-triggered averaging of SNA revealed no correlation with autonomic activity. However, with only 4 neurons confirmed as putative sympathetic premotor neurons vs. 3 neurons in which no autonomic-like behaviour was evident, statistical comparisons are a bit meaningless.

18. p. 24: "Viral anterograde tracing from cl-dSC neurons defined two main descending pathways - a major ventral pathway to the GiA region in the ventromedial medulla, and a minor dorsal pathway that targeted the dorsomedial spinal trigeminal nucleus." In the Results section, the authors identify putative projections of the dSC to the locus coeruleus,

parabrachial/Kolliker-Fuse regions, GiA dorsal, lateral paragigantocellular reticular nucleus, GiA ventral, raphe obscurus, raphe pallidus and raphe magnus. In the legend to Fig 5, it is additionally noted that the dorosommedial spinal trigeminal nucleus is innervated by the dSC. Thus, it is very confusing that in the Discussion, the authors condense the dSC projections into only two pathways, especially considering that the key physiologic experiment to justify their assertion that the GiA is a key relay to mediate autonomic and respiratory effects of dSC activation---the activation of dSC ChR2+ fibers in the GiA---did not fully replicate the effect of dSC light activation.

Author Response: *We thank the Reviewer for this point; we apologise for this confusion. We have clarified the sentence, clarifying that the two bundles identified relate to the medulla, and note that we acknowledge anterograde labelling of the parabrachial/KF complex and PAG as possible mediators of respiratory effects later in the Discussion (P.26).*

Author Response: *Resolved.*

19. p. 25: "intersectional viral tracing failed to identify any other potential relay nuclei between the cl-dSC and spinal cord which may be alternative pathways for the short latency sympathetic responses observed."

Only one section from one animal is shown in Fig 5F to justify this statement. However, even in that single section, there is clearly GFP+ cells beyond the narrow borders of the GiA. These data should be more thoroughly reported to justify the authors' conclusion.

Author Response: *The Reviewer correctly points out that a small number of GFP+ neurons depicted in Figure 5F fall dorsal to the strict boundary of the GiA, placing them within ventromedial compartment of the greater Gigantocellular Reticular Nucleus. However, the take-home message is that labelled neurons identified using this approach accord very closely with the distribution of anterograde labelled dSC projections shown in other parts of the Panel 5 and are certainly concentrated in GiA. We trust that readers of the paper will recognise that the distribution of particular types of neurons is probabilistic rather than absolute with respect to neuroanatomical boundaries, and that this is particularly the case within the reticular formation.*

Action Taken: *Resolved: we have adapted the Results section so that it is less absolute with regard to the result of this experiment, clarifying that GFP+ neurons were concentrated rather than confined within the GiA.*

20. p. 25: "Although previous studies have described a sparse innervation of the rostral ventrolateral medulla (RVLM) by neurons in the SC (Stornetta et al., 2016; Dempsey et al., 2017), anterograde labelling did not reveal anatomical evidence of direct innervation of the RVLM by cl-dSC output neurons in the present study,..." The only data presented to justify this statement is Fig. 5Biv, which appears near the level of the RVLM (it is difficult to be certain without a Nissl counterstain), with positive signal in its ventrolateral extents. Moreover, the legend states "Labelled fibres were infrequently encountered in

the caudal medulla (Biv)." These contradictory data and descriptions of the results are incompatible with their interpretation in the Discussion. Clarification with a more thorough presentation of this difficult experiment would benefit the manuscript.

Author Response: *It's difficult to prove a negative, and we agree that there is some signal around the RVLM region in Figures Biii and Biv. However, we spent many hours searching for boutons on putative RVLM sympathetic premotor neurons identified by their TH immunoreactivity or spinal CTB transport and never found anything convincing. We also hunted for boutons on serotonergic neurons in the raphe nuclei and medial reticular formation, on putative respiratory neurons in the preBotzinger Complex, identified by NK1R immunoreactivity. We conclude that the labelling in the medial RVLM region probably relates to fibres of passage en route to other targets.*

Action Taken: *We agree that these negative findings deserve expansion – we have included a paragraph describing them to the Results.*

21. p. 27: "The results of the present study suggest that the behavioural, autonomic and respiratory responses generated by the caudolateral superior colliculus in response to external salient stimuli are mediated, at least in part, by direct descending inputs to select brainstem nuclei, and do not depend upon connections with forebrain regions" The conclusion that the "behavioural, autonomic and respiratory responses" depend on "direct descending inputs to select brainstem nuclei" is confusing given that the study is entirely focused on whether the GiA mediates autonomic and respiratory responses.

Author Response: *We agree with the reviewer's comment.*

Action Taken: *We have removed the reference to behavioural responses in this statement, so that it now reads "The results of the present study suggest that the autonomic and respiratory responses generated by the caudolateral superior colliculus"*

Minor Concerns

22. p.15-16: The description of the various data analyses made are out of order with respect to the presentation of the data in the Figures and Results sections. It would be helpful to organize these sections in the same order to aid in understanding what was actually done.

Author Response: *We have organised the text detailing the data analysis to be in the same order Figures and Results. Specifically, (1) orientating and behavioural changes to dSC stimulation; (2) Respiratory, autonomic and EEG responses to dSC stimulation in awake rats (4) Effects of dSC stimulation under anaesthesia; (5) Effects of dSC stimulation on GiA unit activity.*

23. Fig. 6, legend: "example illustrating cannula position" should be "[schematic] illustrating cannula position"

Author Response: *The RHS panel of this figure shows a histological specimen (beside the schematic) illustrating ChR2-YFP terminal labelling (white), with anatomical regions overdrawn and the margins of the fibre optic track highlighted.*

Action Taken: *Clarified in the Figure Legend.*

24. Fig. 8D, legend: Bottom panel: What is blue? What is gray? Perhaps separating the PSTH from the raster plot would yield a more legible figure?

Author Response: *Thank you for pointing this out.*

Action Taken: *We have amended the Figure legend to more clearly identify the key features....” D. Low-frequency cl-dSC stimulation evoked short-latency response in the same neuron (laser-triggered peristimulus time histogram (**dark blue**) with overlaid raster (**white spikes, grey background, lower panel**) that preceded simultaneously recorded splanchnic sympathetic responses (green trace, upper panel).”*

25. p.16: What was the concentration of hexamethonium bromide used to determine the noise level in SNA recordings? Its dosage is currently listed as 5mg, which is unclear.

Author Response: *We thank the Reviewer for pointing out this detail for clarification. We administered 5 mg per animal iv, which is the equivalent of ~10 mg/kg. This is an effective dose for complete ganglionic blockade. We have used 8 mg/kg in past papers, and provide a reference to a recent paper of ours that uses this method (Hex iv) for normalisation of SNA.*

Action Taken: *We have amended the methods text to the following and provide a reference to “SNA was normalized relative to baseline (100%) and noise (0%: obtained by hexamethonium bromide at the conclusion of experiments: 5 mg i.v (~10 mg/kg; Sigma Aldrich, Australia), as previously described (Underwood *et al.*, 2022).*

30 August 2022

RE: SPECIAL CASE RESUBMISSION FOR JP-RP-2022-283492

Dear Professor Schultz,

Thank you for providing us with the opportunity to resubmit our paper, *Descending pathways from the superior colliculus mediating autonomic and respiratory effects associated with orienting behaviour*, for consideration at the Journal of Physiology.

We thank the Reviewers for their constructive appraisal of our work, which was rigorous and extremely detailed. We have engaged with the spirit of their feedback to the best of our abilities and provide a substantially revised manuscript.

Reviewer 1 noted the novelty of the research question, the high technical standards of the experiments, and the appropriateness of our interpretation and its broader context, but urged some caution regarding identification of putative presympathetic neurons, which is valid and helpful.

Reviewer 2 did not articulate any specific concerns regarding the substance of our work or its broader context in the field, but made a considerable number of recommendations and queries regarding the representation of the data. While we do not agree with every point made (there were 25!), we have endeavoured to explain our choices in a detailed point-by-point rebuttal, which includes alternative representations of the data for their consideration.

Taken together, the Reviewers' comments have provided helpful insights into areas of the manuscript that can be improved and limitations which have now been addressed in the text. We hope that you and they feel it now suitable for publication in the Journal of Physiology.

Given the paper is to be sent out again for review, we have not yet completed the statistical summary or supplied a first author profile, but will of course happily provide this additional information upon provisional acceptance of the paper.

Yours sincerely

Simon McMullan

Interim Co-Head, Macquarie Medical School
Associate Professor & Group Leader, Neurobiology of Vital Systems
Macquarie University
NSW 2109 Australia

E: simon.mcmullan@mq.edu.au
T: +61 (2) 9850 2710

Dear Dr McMullan,

Re: JP-RP-2022-283789X "Descending pathways from the superior colliculus mediating autonomic and respiratory effects associated with orienting behaviour" by Erin Lynch, Bowen R Dempsey, Christine Saleeba, Eloise Monteiro, Anita Turner, Peter GR Burke, Andrew M Allen, Roger Dampney, Cara Margaret Hildreth, Jennifer Cornish, Ann K Goodchild, and Simon McMullan

Thank you for resubmitting your revised Research Article to The Journal of Physiology. It has been assessed by the original Reviewing Editor and Referees and has been well received. Some final revisions have been requested.

The reports are copied at the end of this email. Please address all of the points and incorporate all requested revisions, or explain in your Response to Referees why a change has not been made.

NEW POLICY: In order to improve the transparency of its peer review process The Journal of Physiology publishes online as supporting information the peer review history of all articles accepted for publication. Readers will have access to decision letters, including all Editors' comments and referee reports, for each version of the manuscript and any author responses to peer review comments. Referees can decide whether or not they wish to be named on the peer review history document.

Authors are asked to use The Journal's premium BioRender (<https://biorender.com/>) account to create/redraw their Abstract Figures. Information on how to access The Journal's premium BioRender account is here: <https://physoc.onlinelibrary.wiley.com/journal/14697793/biorender-access> and authors are expected to use this service. This will enable Authors to download high-resolution versions of their figures. The link provided should only be used for the purposes of this submission. Authors will be charged for figures created on this premium BioRender account if they are not related to this manuscript submission.

I hope you will find the comments helpful and have no difficulty returning your revisions within 4 weeks.

Your revised manuscript should be submitted online using the links in Author Tasks Link Not Available.

Any image files uploaded with the previous version are retained on the system. Please ensure you replace or remove all files that have been revised.

REVISION CHECKLIST:

- Article file, including any tables and figure legends, must be in an editable format (eg Word)
- Abstract figure file (see above)
- Statistical Summary Document
- Upload each figure as a separate high quality file
- Upload a full Response to Referees, including a response to any Senior and Reviewing Editor Comments;
- Upload a copy of the manuscript with the changes highlighted.

- A potential 'Cover Art' file for consideration as the Issue's cover image;
- Appropriate Supporting Information (Video, audio or data set https://jp.msubmit.net/cgi-bin/main.plex?form_type=display_requirements#supp).

To create your 'Response to Referees' copy all the reports, including any comments from the Senior and Reviewing Editors, into a Word, or similar, file and respond to each point in colour or CAPITALS and upload this when you submit your revision.

I look forward to receiving your revised submission.

If you have any queries please reply to this email and staff will be happy to assist.

Yours sincerely,

Harold D Schultz
Senior Editor
The Journal of Physiology
<https://jp.msubmit.net>
<http://jp.physoc.org>
The Physiological Society
Hodgkin Huxley House
30 Farringdon Lane
London, EC1R 3AW
UK
<http://www.physoc.org>
<http://journals.physoc.org>

REQUIRED ITEMS FOR REVISION:

-You must start the Methods section with a paragraph headed Ethical Approval. A detailed explanation of journal policy and regulations on animal experimentation is given in Principles and standards for reporting animal experiments in The Journal of Physiology and Experimental Physiology by David Grundy J Physiol, 593: 2547-2549. doi:10.1113/JP270818.). A checklist outlining these requirements and detailing the information that must be provided in the paper can be found at: <https://physoc.onlinelibrary.wiley.com/hub/animal-experiments>. Authors should confirm in their Methods section that their experiments were carried out according to the guidelines laid down by their institution's animal welfare committee, and conform to the principles and regulations as described in the Editorial by Grundy (2015). The Methods section must contain details of the anaesthetic regime: anaesthetic used, dose and route of administration and method of killing the experimental animals.

-Your manuscript must include a complete Additional Information section

-The Journal of Physiology funds authors of provisionally accepted papers to use the premium BioRender site to create high resolution schematic figures. Follow this link and enter your details and the manuscript number to create and download figures. Upload these as the figure files for your revised submission. If you choose not to take up this offer we require figures to be of similar quality and resolution. If you are opting out of this service to authors, state this in the Comments section on the Detailed Information page of the submission form. The link provided should only be used for the purposes of this submission. Authors will be charged for figures created on this premium BioRender account if they are not related to this manuscript submission.

-Please upload separate high-quality figure files via the submission form.

-A Statistical Summary Document, summarising the statistics presented in the manuscript, is required upon revision. It must be on the Journal's template, which can be downloaded from the link in the Statistical Summary Document section here: https://jp.msubmit.net/cgi-bin/main.plex?form_type=display_requirements#statistics

-Papers must comply with the Statistics Policy https://jp.msubmit.net/cgi-bin/main.plex?form_type=display_requirements#statistics

In summary:

-If n {less than or equal to} 30, all data points must be plotted in the figure in a way that reveals their range and distribution. A bar graph with data points overlaid, a box and whisker plot or a violin plot (preferably with data points included) are acceptable formats.

-If $n > 30$, then the entire raw dataset must be made available either as supporting information, or hosted on a not-for-profit repository e.g. FigShare, with access details provided in the manuscript.

- n clearly defined (e.g. x cells from y slices in z animals) in the Methods. Authors should be mindful of pseudoreplication.

- All relevant 'n' values must be clearly stated in the main text, figures and tables, and the Statistical Summary Document (required upon revision)
- The most appropriate summary statistic (e.g. mean or median and standard deviation) must be used. Standard Error of the Mean (SEM) alone is not permitted.
- Exact p values must be stated. Authors must not use 'greater than' or 'less than'. Exact p values must be stated to three significant figures even when 'no statistical significance' is claimed.
- Statistics Summary Document completed appropriately upon revision

-A Data Availability Statement is required for all papers reporting original data. This must be in the Additional Information section of the manuscript itself. It must have the paragraph heading "Data Availability Statement". All data supporting the results in the paper must be either: in the paper itself; uploaded as Supporting Information for Online Publication; or archived in an appropriate public repository. The statement needs to describe the availability or the absence of shared data. Authors must include in their Statement: a link to the repository they have used, or a statement that it is available as Supporting Information; reference the data in the appropriate section(s) of their manuscript; and cite the data they have shared in the References section. Whenever possible the scripts and other artefacts used to generate the analyses presented in the paper should also be publicly archived. If sharing data compromises ethical standards or legal requirements then authors are not expected to share it, but must note this in their Statement. For more information, see our Statistics Policy.

-Please include an Abstract Figure. The Abstract Figure is a piece of artwork designed to give readers an immediate understanding of the research and should summarise the main conclusions. If possible, the image should be easily 'readable' from left to right or top to bottom. It should show the physiological relevance of the manuscript so readers can assess the importance and content of its findings. Abstract Figures should not merely recapitulate other figures in the manuscript. Please try to keep the diagram as simple as possible and without superfluous information that may distract from the main conclusion(s). Abstract Figures must be provided by authors no later than the revised manuscript stage and should be uploaded as a separate file during online submission labelled as File Type 'Abstract Figure'. Please ensure that you include the figure legend in the main article file. All Abstract Figures should be created using BioRender. Authors should use The Journal's premium BioRender account to export high-resolution images. Details on how to use and access the premium account are included as part of this email.

EDITOR COMMENTS

Reviewing Editor:

Thank you for addressing the Reviewers' comments, especially the very detailed comments provided by Reviewer 2, who is very satisfied with your responses. As you will see, only minor concerns have been raised by Reviewer 1, who prefers that you be more circumspect in your conclusions. Indeed, in comments to me, this Reviewer believes that your evidence is insufficient to dismiss a possible contribution of the RVLM to the cardiovascular responses elicited by stimulation of the superior colliculus, going on to suggest that this possibility be acknowledged unless you can produce more compelling evidence to the contrary.

Senior Editor:

Comments for Authors to ensure the paper complies with the Statistics Policy:
Please include actual p values throughout (unless $P < .0001$), including within the figures (Fig. 1; Fig. 2; Fig. 3; Fig. 4; Fig. 7). Ensure that samples sizes and statistical test(s) used are included in the legends for these figures.

If the statistical summary document has errors please describe what is incorrect:
The statistical summary document must be provided with the re-submission.

Comments to the Author:

The manuscript has been approved with only minor comments remain to be addressed from reviewers. In addition, the manuscript must still address formatting requirements as addressed in Instructions to Authors. The first paragraph of the Methods should be labeled Ethical Approval and contain a statement that the investigators understand the ethical principles under which the journal operates and that their work complies with the animal ethics as outlined by the journal.

(<https://physoc.onlinelibrary.wiley.com/hub/animal-experiments>). All surgical procedures must provide details of anesthesia (premedication, drugs, dosages, route, supplementation, method of assessing plane of anesthesia throughout), post surgical care to minimize pain and suffering (analgesia, antibiotics, recovery period), and euthanasia (drug, dose, confirmation of death). Please comply with the journal policy on reporting statistics (https://jp.msubmit.net/cgi-bin/main.plex?form_type=display_requirements#statistics). Please include actual p values throughout (unless $P < .0001$), including in the figures when possible (Fig. 1; Fig. 2; Fig. 3; Fig. 4; Fig. 7). Ensure that samples sizes and statistical test(s) used are included in the legends for these figures.

REFEREE COMMENTS

Referee #1:

My prior concerns were minor and nicely addressed. The following remark may still be worth considering.

Page 26: "other studies have shown that they are not directly responsible for mediating sympathetic responses evoked by psychological stress (Carrive & Gorissen, 2008; Furlong et al., 2014). Consistent with these previous findings, our results indicate that RVLM sympathetic premotor neurons do not generate sympathetic responses evoked by external salient stimuli."

The authors may want to say "have suggested" rather than "have shown". This negative evidence was apparently based on the absence of an uptick in Fos expression in the C1 neurons following the behavioral challenges. Fos is a very insensitive index of neuronal activation. This negative evidence does not exclude the possibility that these neurons might have been notably activated, although admittedly not to the extent produced by severe physical challenges such as sustained hypotension, hypoxia or hemorrhage.

The second sentence also seems too assertive. Later in the discussion the authors note, appropriately, that the RVLM presympathetic neurons could have been activated indirectly (e.g. via additional interneurons, collaterals of the bulbospinal GiA neurons?). The authors did not record from RVLM presympathetic neurons (C1 and others) in the present study and thus cannot really exclude the possibility that these neurons could be activated. Also Why assume a priori that the autonomic responses evoked by SC stimulation could not be mediated by the combined action of multiple descending pre-autonomic neuronal populations?

Referee #2:

The manuscript elucidates a pathway that mediates the physiologic components of the orienting response. The strong experimental approach and significance of the question will be impactful for researchers in a variety of fields. The authors have thoroughly addressed my concerns and significantly improved the manuscript. I have no further questions or concerns.

END OF COMMENTS

1st Confidential Review

30-Aug-2022

MACQUARIE MEDICAL SCHOOL
Faculty of Medicine, Health & Human Sciences
F10A Macquarie University
NSW 2109

11 October 2022

Harold D Schultz
Senior Editor
The Journal of Physiology
Hodgkin Huxley House
30 Farringdon Lane
London, EC1R 3AW
UK

RE: JP-RP-2022-283789XR1

Dear Dr Schultz,

Thank you for the opportunity to resubmit our paper – we have implemented all of the recommended changes (as described below) and confirm conformation to the Journal's standards for the reporting of statistical data and work on experimental animals. We hope you find it now ready for publication.

Yours sincerely

Simon McMullan

Interim Co-Head, Macquarie Medical School
Associate Professor & Group Leader, Neurobiology of Vital Systems
Macquarie University
NSW 2109 Australia

E: simon.mcmullan@mq.edu.au
T: +61 (2) 9850 2710

EDITOR COMMENTS

Reviewing Editor:

Thank you for addressing the Reviewers' comments, especially the very detailed comments provided by Reviewer 2, who is very satisfied with your responses. As you will see, only minor concerns have been raised by Reviewer 1, who prefers that you be more circumspect in your conclusions. Indeed, in comments to me, this Reviewer believes that your evidence is insufficient to dismiss a possible contribution of the RVLM to the cardiovascular responses elicited by stimulation of the superior colliculus, going on to suggest that this possibility be acknowledged unless you can produce more compelling evidence to the contrary.

Author Response: Thank you – we have toned down our interpretation of the issue regarding involvement of RVLM neurons and agree that there is, at present, insufficient evidence to discount their involvement.

Senior Editor:

Comments for Authors to ensure the paper complies with the Statistics Policy:

Please include actual p values throughout (unless $P < ,0001$), including within the figures (Fig. 1; Fig. 2; Fig. 3; Fig. 4; Fig. 7). Ensure that samples sizes and statistical test(s) used are included in the legends for these figures.

If the statistical summary document has errors please describe what is incorrect:

The statistical summary document must be provided with the re-submission.

Author Response: I confirm that these data are included in the figures and summary document.

Comments to the Author:

The manuscript has been approved with only minor comments remain to be addressed from reviewers. In addition, the manuscript must still address formatting requirements as addressed in Instructions to Authors. The first paragraph of the Methods should be labeled Ethical Approval and contain a statement that the investigators understand the ethical principles under which the journal operates and that their work complies with the animal ethics as outlined by the journal. (<https://physoc.onlinelibrary.wiley.com/hub/animal-experiments>). All surgical procedures must provide details of anesthesia (premedication, drugs, dosages, route, supplementation, method of assessing plane of anesthesia throughout), post surgical care to minimize pain and suffering (analgesia, antibiotics, recovery period), and euthanasia (drug, dose, confirmation of death). Please comply with the journal policy on reporting statistics (https://jp.msubmit.net/cgi-bin/main.plex?form_type=display_requirements#statistics). Please include actual p values throughout (unless $P < ,0001$), including in the figures when possible (Fig. 1; Fig. 2; Fig. 3; Fig. 4; Fig. 7). Ensure that samples sizes and statistical test(s) used are included in the legends for these figures.

Author Response: I confirm that the ethical statement and requested information have been included in the revised manuscript.

REFeree COMMENTS

Referee #1:

My prior concerns were minor and nicely addressed. The following remark may still be worth considering.

Author Response: Thank you for the additional feedback for our discussion points and we agree wholeheartedly with the issues raised below relating to the network(s) of pre-autonomic neuronal populations that could mediate the sympathetic response to dSC stimulation. We don't have evidence to exclude the RVLM or A5 neurons as additional networks contributing to the sympathetic activation and we have now tempered these discussion points. Thank you for these suggestions.

Page 26: "other studies have shown that they are not directly responsible for mediating sympathetic responses evoked by psychological stress (Carrive & Gorissen, 2008; Furlong et al., 2014). Consistent with these previous findings, our results indicate that RVLM sympathetic premotor neurons do not generate sympathetic responses evoked by external salient stimuli."

The authors may want to say "have suggested" rather than "have shown". This negative evidence was apparently based on the absence of an uptick in Fos expression in the C1 neurons following the behavioral challenges. Fos is a very insensitive index of neuronal activation. This negative evidence does not exclude the possibility that these neurons might have been notably activated, although admittedly not to the extent produced by severe physical challenges such as sustained hypotension, hypoxia or hemorrhage.

The second sentence also seems too assertive. Later in the discussion the authors note, appropriately, that the RVLM presympathetic neurons could have been activated indirectly (e.g. via additional interneurons, collaterals of the bulbospinal GiA neurons?). The authors did not record from RVLM presympathetic neurons (C1 and others) in the present study and thus cannot really exclude the possibility that these neurons could be activated. Also Why assume a priori that the autonomic responses evoked by SC stimulation could not be mediated by the combined action of multiple descending pre-autonomic neuronal populations?

Author Response: We agree with this point – we don't have the evidence to exclude other pre-autonomic networks. We have eliminated the assertive language, though we do maintain the argument that the sympathetic effects are most likely via GiA. We hope our amendments reflect the possibility for multiple descending pathways, if not via GiA. Thank you for the critique of our study and discussion points.

Referee #2:

The manuscript elucidates a pathway that mediates the physiologic components of the orienting response. The strong experimental approach and significance of the question will be impactful for researchers in a variety of fields. The authors have thoroughly addressed my concerns and significantly improved the manuscript. I have no further questions or concerns.

Author Response: Thank you for this endorsement – we appreciate your time and feedback on our study

Dear Dr McMullan,

Re: JP-RP-2022-283789XR1 "Descending pathways from the superior colliculus mediating autonomic and respiratory effects associated with orienting behaviour" by Erin Lynch, Bowen R Dempsey, Christine Saleeba, Eloise Monteiro, Anita Turner, Peter GR Burke, Andrew M Allen, Roger Dampney, Cara Margaret Hildreth, Jennifer Cornish, Ann K Goodchild, and Simon McMullan

I am pleased to tell you that your paper has been accepted for publication in The Journal of Physiology.

NEW POLICY: In order to improve the transparency of its peer review process, The Journal of Physiology publishes online as supporting information the peer review history of all articles accepted for publication. Readers will have access to decision letters, including all Editors' comments and referee reports, for each version of the manuscript and any author responses to peer review comments. Referees can decide whether or not they wish to be named on the peer review history document.

The last Word version of the paper submitted will be used by the Production Editors to prepare your proof. When this is ready you will receive an email containing a link to Wiley's Online Proofing System. The proof should be checked and corrected as quickly as possible.

Authors should note that it is too late at this point to offer corrections prior to proofing. The accepted version will be published online, ahead of the copy edited and typeset version being made available. Major corrections at proof stage, such as changes to figures, will be referred to the Reviewing Editor for approval before they can be incorporated. Only minor changes, such as to style and consistency, should be made a proof stage. Changes that need to be made after proof stage will usually require a formal correction notice.

All queries at proof stage should be sent to TJP@wiley.com.

Are you on Twitter? Once your paper is online, why not share your achievement with your followers. Please tag The Journal (@jphysiol) in any tweets and we will share your accepted paper with our 23,000+ followers!

Yours sincerely,

Harold D Schultz
Senior Editor
The Journal of Physiology
<https://jp.msubmit.net>
<http://jp.physoc.org>
The Physiological Society
Hodgkin Huxley House
30 Farringdon Lane
London, EC1R 3AW
UK
<http://www.physoc.org>
<http://journals.physoc.org>

P.S. - You can help your research get the attention it deserves! Check out Wiley's free Promotion Guide for best-practice recommendations for promoting your work at www.wileyauthors.com/eeo/guide. And learn more about Wiley Editing Services which offers professional video, design, and writing services to create shareable video abstracts, infographics, conference posters, lay summaries, and research news stories for your research at www.wileyauthors.com/eeo/promotion.

*** IMPORTANT NOTICE ABOUT OPEN ACCESS ***

To assist authors whose funding agencies mandate public access to published research findings sooner than 12 months after publication The Journal of Physiology allows authors to pay an open access (OA) fee to have their papers made freely available immediately on publication.

You will receive an email from Wiley with details on how to register or log-in to Wiley Authors Services where you will be able to place an OnlineOpen order.

You can check if your funder or institution has a Wiley Open Access Account here <https://authorservices.wiley.com/author-resources/Journal-Authors/licensing-and-open-access/open-access/author-compliance-tool.html>

Your article will be made Open Access upon publication, or as soon as payment is received.

If you wish to put your paper on an OA website such as PMC or UKPMC or your institutional repository within 12 months of publication you must pay the open access fee, which covers the cost of publication.

OnlineOpen articles are deposited in PubMed Central (PMC) and PMC mirror sites. Authors of OnlineOpen articles are permitted to post the final, published PDF of their article on a website, institutional repository, or other free public server, immediately on publication.

Note to NIH-funded authors: The Journal of Physiology is published on PMC 12 months after publication, NIH-funded authors DO NOT NEED to pay to publish and DO NOT NEED to post their accepted papers on PMC.

EDITOR COMMENTS

Reviewing Editor:

Thank you for attending to this remaining comment from Reviewer 1, which I believe you have done satisfactorily.

Senior Editor:

Thank you for submitting your manuscript to The Journal of Physiology. We are very pleased to accept your paper for publication. Please consider The Journal for your future work.